# Decoupling from yolk sac is required for extraembryonic tissue spreading in the scuttle fly *Megaselia abdita*

Francesca Caroti[1†‡], Everardo González Avalos[1†], Viola Noeske[1†], Paula González Avalos[1], Dimitri Kromm[2,3], Maike Wosch[1], Lucas Schütz[1], Lars Hufnagel[2], Steffen Lemke[1]*

[1]Centre for Organismal Studies Heidelberg, Heidelberg, Germany; [2]European Molecular Biology Laboratory, Heidelberg, Germany; [3]Collaboration for joint PhD degree between EMBL and Heidelberg University, Faculty of Biosciences, Heidelberg University, Heidelberg, Germany

**\*For correspondence:**
steffen.lemke@cos.uni-heidelberg.de

[†]These authors contributed equally to this work

**Present address:** [‡]Centre of Microbial and Plant Genetics, Leuven, Belgium

**Competing interests:** The authors declare that no competing interests exist.

**Abstract** Extraembryonic tissues contribute to animal development, which often entails spreading over embryo or yolk. Apart from changes in cell shape, the requirements for this tissue spreading are not well understood. Here, we analyze spreading of the extraembryonic serosa in the scuttle fly *Megaselia abdita*. The serosa forms from a columnar blastoderm anlage, becomes a squamous epithelium, and eventually spreads over the embryo proper. We describe the dynamics of this process in long-term, whole-embryo time-lapse recordings, demonstrating that free serosa spreading is preceded by a prolonged pause in tissue expansion. Closer examination of this pause reveals mechanical coupling to the underlying yolk sac, which is later released. We find mechanical coupling prolonged and serosa spreading impaired after knockdown of *M. abdita Matrix metalloprotease 1*. We conclude that tissue–tissue interactions provide a critical functional element to constrain spreading epithelia.
DOI: https://doi.org/10.7554/eLife.34616.001

## Introduction

In the early stages of animal development, pre-patterned cells are collectively set aside from the embryo proper to differentiate into specialized, extraembryonic epithelia, which then generate a local environment and support the growing organism from outside the embryo (*Wolpert and Tickle, 2011*). In chick, quails, frogs, fish and other vertebrates with a yolk-rich egg, such extraembryonic epithelia typically expand from the periphery of the embryo, spread over, and eventually envelope the underlying yolk cell (*Downie, 1976*; *Futterman et al., 2011*; *Keller, 1980*; *Trinkaus, 1951*; *Arendt and Nübler-Jung, 1999*). In bugs, beetles, butterflies, mosquitoes and other insects, extraembryonic epithelia already develop on top of a yolk sac, and when they spread, they eventually envelope the embryo proper (*Panfilio et al., 2006*; *Handel et al., 2000*; *Kraft and Jäckle, 1994*; *Goltsev et al., 2009*; *Anderson, 1972a*; *Anderson, 1972b*). To form such epithelial envelopes, extraembryonic tissues dramatically expand their area and often undergo a transformation from a columnar architecture with thin and tall cells to a squamous epithelium with spread-out and short cells (*Anderson, 1972a*; *Anderson, 1972b*; *Bruce, 2016*).

While synchronized and cell-autonomous flattening of cells is likely a contributing component (*Pope and Harris, 2008*), work from the fruit fly *Drosophila melanogaster* suggests that cell shape changes alone may not be sufficient to explain extraembryonic tissue spreading in insects (*Lacy and Hutson, 2016*). In *D. melanogaster*, cells of the extraembryonic amnioserosa synchronously and autonomously change their shape and thereby transform an initially columnar tissue into a squamous

epithelium (*Pope and Harris, 2008*; *Campos-Ortega and Hartenstein, 1997*). In contrast to the extraembryonic tissues in most other insects, however, the amnioserosa does not extend from its dorsal position over the yolk and never spreads over the embryo (*Panfilio, 2008*; *Schmidt-Ott and Kwan, 2016*). These observations suggest that additional cellular mechanisms are required to instruct or permit extraembryonic tissue spreading.

To identify unknown mechanisms required for extraembryonic tissue spreading in insects, we investigated early development in the scuttle fly *Megaselia abdita*. *M. abdita* shared a last common ancestor with *D. melanogaster* about 150 million years ago (*Wiegmann et al., 2011*). While overall embryonic development of *M. abdita* and *D. melanogaster* is conserved and comparable (*Wotton et al., 2014*), the two species differ in their extraembryonic development (*Rafiqi et al., 2008*; *Wotton et al., 2014*). In *D. melanogaster*, the amnioserosa is set up as a single extraembryonic epithelium by expression of the Hox3 transcription factor Zerknüllt (Zen), which is controlled by peak levels of BMP signaling along the dorsal midline of the blastoderm embryo (*Gavin-Smyth and Ferguson, 2014*). In contrast to *D. melanogaster*, *M. abdita* develops two distinct extraembryonic tissues, the serosa and, bordering it, the amnion (*Kwan et al., 2016*; *Rafiqi et al., 2008*; *Rafiqi et al., 2012*). In the *M. abdita* blastoderm embryo, expression of the *zen* orthologue defines the serosa anlage along the dorsal midline (*Rafiqi et al., 2008*). Adjacent to this *zen*-defined serosa anlage, but within the range of dorsal BMP signaling, a narrow domain void of both *zen* and general embryonic patterning genes was identified as putative amnion anlage (*Figure 1A*; (*Kwan et al., 2016*; *Rafiqi et al., 2012*)). Cells of the serosa and amnion anlage undergo synchronous cell shape changes and eventually differentiate into squamous epithelia. The serosa then separates from the adjacent amnion, spreads freely, and continuously increases its cell size until it closes as perfect envelope on the ventral side of the egg (*Rafiqi et al., 2008*; *Rafiqi et al., 2010*). Because the serosa in *M. abdita* has retained the ancestral ability to expand and envelope the embryo proper, the species has been previously identified as key organism to understand the evolutionary origin of the amnioserosa as a single, non-spreading extraembryonic epithelium (*Hallgrímsson et al., 2012*).

To take advantage of its close relationship to *D. melanogaster* and use it as model to address cell-biological mechanisms of extraembryonic tissue spreading, tissue and cell dynamics in *M. abdita* development are required to be studied with high spatiotemporal resolution. Here we have established time-lapse recordings at the necessary resolution in injected embryos using confocal and light sheet microscopy. We identified mechanical coupling between serosa and yolk sac as a critical element to control serosa spreading and found that changes in tissue-tissue interaction provide a compelling variable for the evolution of epithelial spreading.

## Results

### *In toto* time-lapse recordings faithfully recover known landmarks of embryonic and extraembryonic development in *M. abdita*

Extraembryonic development in *M. abdita* has been characterized by the formation of two extraembryonic tissues, the amnion and the serosa. Both tissues develop from columnar cells of the blastoderm embryo and have been associated with dramatic changes in cell size. To test whether such cellular properties could be used to trace amnion and serosa differentiation from the blastoderm stage onwards, we carefully examined fixed specimen by sampling subsequent time points of development using precisely staged depositions (*Figure 1—figure supplement 1*). Our quantitative analyses of cell shapes clearly showed two classes of cells, that is large and flat and presumably extraembryonic cells, and small and round and presumably ectodermal cells (*Figure 1—figure supplement 1*). However, our data were not sufficient to distinguish between putative amnion and serosa cells, which made it impossible to follow their development from the blastoderm embryo.

Time-lapse recordings of individual living specimen permit the tracing of cells and tissues throughout development (*Keller, 2013*), which avoids the discontinuity that limited our analysis of fixed specimen. In case of extraembryonic development in *M. abdita*, this approach would require imaging of the entire embryo (the serosa is specified on the dorsal side and eventually fuses on the ventral side [*Rafiqi et al., 2008*]), at cellular resolution, and for an extended period of time. Time-lapse recordings of embryonic development using selective plane illumination microscopy (SPIM) have been previously shown to provide the spatiotemporal resolution necessary for similar analyses

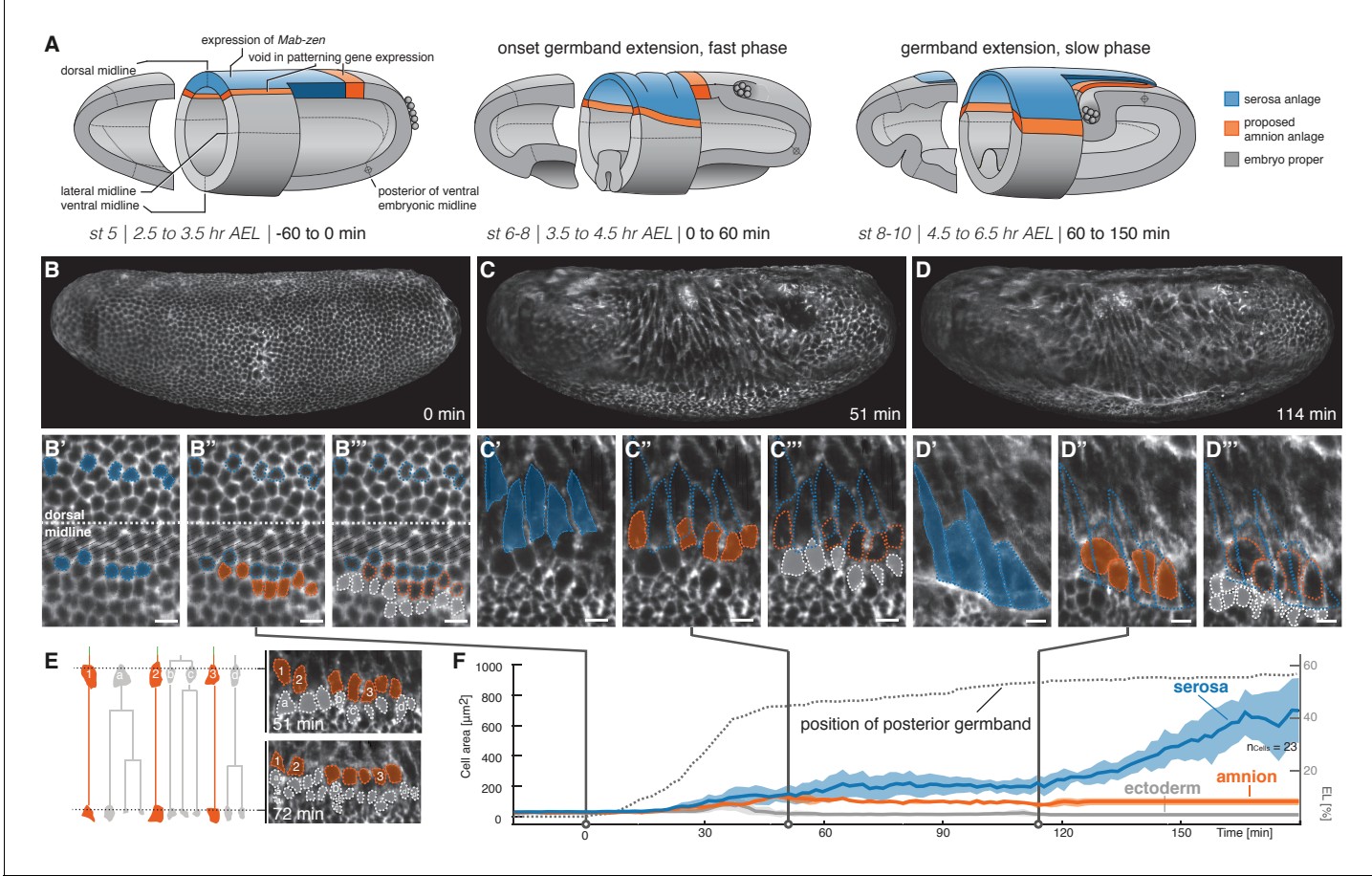

**Figure 1.** Tracking of blastoderm cells characterizes serosa and amnion differentiation. (**A**) Model of early extraembryonic tissue development in *M. abdita* based on marker gene expression in fixed specimen. Stage (st) and time after egg lay (AEL) are defined as in (*Wotton et al., 2014*); absolute time given in minutes relative to the onset of germband extension (onset GBE = 0 min). (**B–D'''**) Global embryonic views of SPIM recorded embryos at corresponding stages (**B–D**), with tracked and marked serosa (blue), amnion (orange), and ectoderm cells (grey) in 2D-projections of indicated surface areas in dorsal (**B'–B''**) and lateral views (**C'–C'''**, **D'–D'''**). Cells of serosa were identified based on their ability to spread over the embryo and then tracked back to the cellular blastoderm. (**E**) Cell lineage and divisions in putative amnion and ectoderm cells. Cells directly adjacent to the serosa never divided and could be back-tracked to a single row of cells next to the serosa anlage; these cells were classified as presumptive amnion. Cells further distal to the serosa divided, eventually decreased in cell size, and were classified as presumptive ectoderm. (**F**) Quantitative analysis of cell size of tracked serosa, amnion, and ectodermal cells relative to GBE as measure of developmental progression. The position of the posterior germband is indicated in % egg length (0% EL = posterior pole; dotted line); Standard error of mean shown as shades. Unless indicated otherwise, embryos and close-ups are shown with anterior left and dorsal up. Scale bars, 10 μm.

DOI: https://doi.org/10.7554/eLife.34616.002

The following figure supplement is available for figure 1:

**Figure supplement 1.** Quantitative analyses of cell measures in *M.abdita* embryos fixed at subsequent stages of development.

DOI: https://doi.org/10.7554/eLife.34616.003

(*Chhetri et al., 2015*; *Rauzi et al., 2015*; *Wolff et al., 2018*), suggesting that SPIM could also be used to explore formation and differentiation of extraembryonic tissues in the *M. abdita* embryo.

To establish SPIM imaging for *M. abdita* embryos, we first established fluorescent reporters for cell outline and nuclei (Lifeact and Histone H1, see Materials and methods). Next, we tested whether overall development was affected by long-term imaging of injected *M. abdita* embryos. To assess the putative impact of extended time-lapse recordings, we compared universal landmarks of fly development (onset of germband extension, onset of germband retraction, and end of germband retraction) in SPIM recordings with our own live observations as well as previously described recordings of *M. abdita* development (*Wotton et al., 2014*). For all three landmarks, our live observations confirmed previously described timing (onset of germband extension 3:45 hr after egg laying (AEL),

onset of germband retraction 8:10 hr AEL, end of germband retraction 11:05 hr AEL [*Wotton et al., 2014*]) within a window of 15 min (n = 24), which corresponded to the time interval of egg deposition. In our quantified SPIM recordings, we found the same timing of events, suggesting that development of injected *M. abdita* embryos was not notably affected by long-term SPIM imaging. In the following, we will use previously defined staging and terminology for *M. abdita* development where possible (*Rafiqi et al., 2008*; *Wotton et al., 2014*), while absolute times will be provided relative to the onset of germband extension (0 min).

To follow development and differentiation of *M. abdita* serosa and amnion from their proposed blastoderm anlage, we selected a representative time-lapse recording for cell tracking and followed cells of putative serosa, amnion, and ectoderm throughout early development (*Figure 1B–D'''*). The serosa developed from a dorsal anlage that was about six to seven cells wide (*Figure 1B,B'*). These cells did not divide, increased in cell surface area (*Figure 1C,C'*), and eventually spread over the adjacent tissue (*Figure 1D,D'*). The width of the inferred serosa domain perfectly coincided with gene expression of the known serosa key regulator, the homeodomain transcription factor Zerknüllt (Zen), while the observed cell dynamics matched serosa behavior as previously inferred from fixed specimen (*Rafiqi et al., 2008*). Directly adjacent to the unambiguously identified serosa anlage, we identified a single row of cells (*Figure 1B,B''*), which increased in apical cell area (*Figure 1C,C''*) but eventually did not become part of the spreading serosa (*Figure 1D,D''*). These cells did not divide (*Figure 1E*), and, consistent with patterning information in the early blastoderm embryo (*Figure 1A*; (*Kwan et al., 2016*; *Rafiqi et al., 2012*)), formed the presumptive amnion. Next to the amnion, cells increased in apical cell area temporarily (*Figure 1B,B''',C,C'''*) but eventually divided and decreased in size (*Figure 1D,D''',E*), suggesting differentiation into ectodermal cells. The overall analyses of surface cell area at later stages of development, that is after germband extension, indicated a substantial increase in surface area for serosa cells (up to 10-fold), a still notable increase for presumptive amnion cells (about 2-fold), and a slight reduction of cell surface area in the ectoderm (*Figure 1F*). Taken together, our cell tracking analyses support the previously proposed model of the *M. abdita* blastoderm fate map and set the stage for a more global analysis of amnion and serosa tissue behavior.

## The amnion in *M. abdita* develops as a lateral stripe of cells

Previous analyses of *M. abdita* amnion development have led to conflicting hypotheses regarding its position and topology. Based on either cell outline or marker gene expression, the *M. abdita* amnion has been suggested to consist of either a dorsally closed epithelium or a thin lateral stripe of cells (*Rafiqi et al., 2008*; *Kwan et al., 2016*). To resolve this question, we aimed to follow expansion and development of the presumptive amnion in our time-lapse recordings. To achieve this in the absence of a specific *in vivo* reporter, we developed a set of image processing routines for embryos recorded in isotropic three dimensional (3D) volumes. These routines allowed us first to identify the surface of the embryo and to digitally 'peel off' individual layers until the serosa was removed. We then flattened the surface into a two dimensional (2D) carpet and marked the remaining cells with large surface area as presumptive amnion. Finally, these carpets were projected again into the initial volume to provide 3D renderings (*Figure 2—figure supplement 1*; Materials and methods).

Thus following development of the presumptive amnion through consecutive stages of development, our results suggested that the *M. abdita* amnion consisted of a lateral tissue, which was essentially one cell wide, and a cap at the posterior end of the germband that closed over the ventral side of the embryo (which, because of the extended germband, faced the dorsal side of the egg; *Figure 2A–D*). During germband extension and up until the onset of germband retraction, cells of the presumptive lateral amnion had a rather smooth outline and appeared to be folded over the ectoderm; with the onset of germband retraction, these cells then seemed to reside more over the yolk and developed notable protrusions towards the dorsal midline of the embryo (*Figure 2C,D*). To follow the position of the presumptive amnion more precisely, we tracked individual amnion cells along the embryonic circumference over the course of germband extension and retraction. Our analyses further consolidated the notion of a lateroventral amnion in *M. abdita* (*Figure 2E,F*).

To understand how amnion cells behaved during and after their separation from the serosa, we computed donut-like sections of the developing embryo that allowed us to observe embryo surface and transverse section in 3D renderings (*Figure 2G*; *Figure 2—video1*). In these renderings, we then analyzed the behavior of lateral amnion cells relative to the serosa by using enrichment of

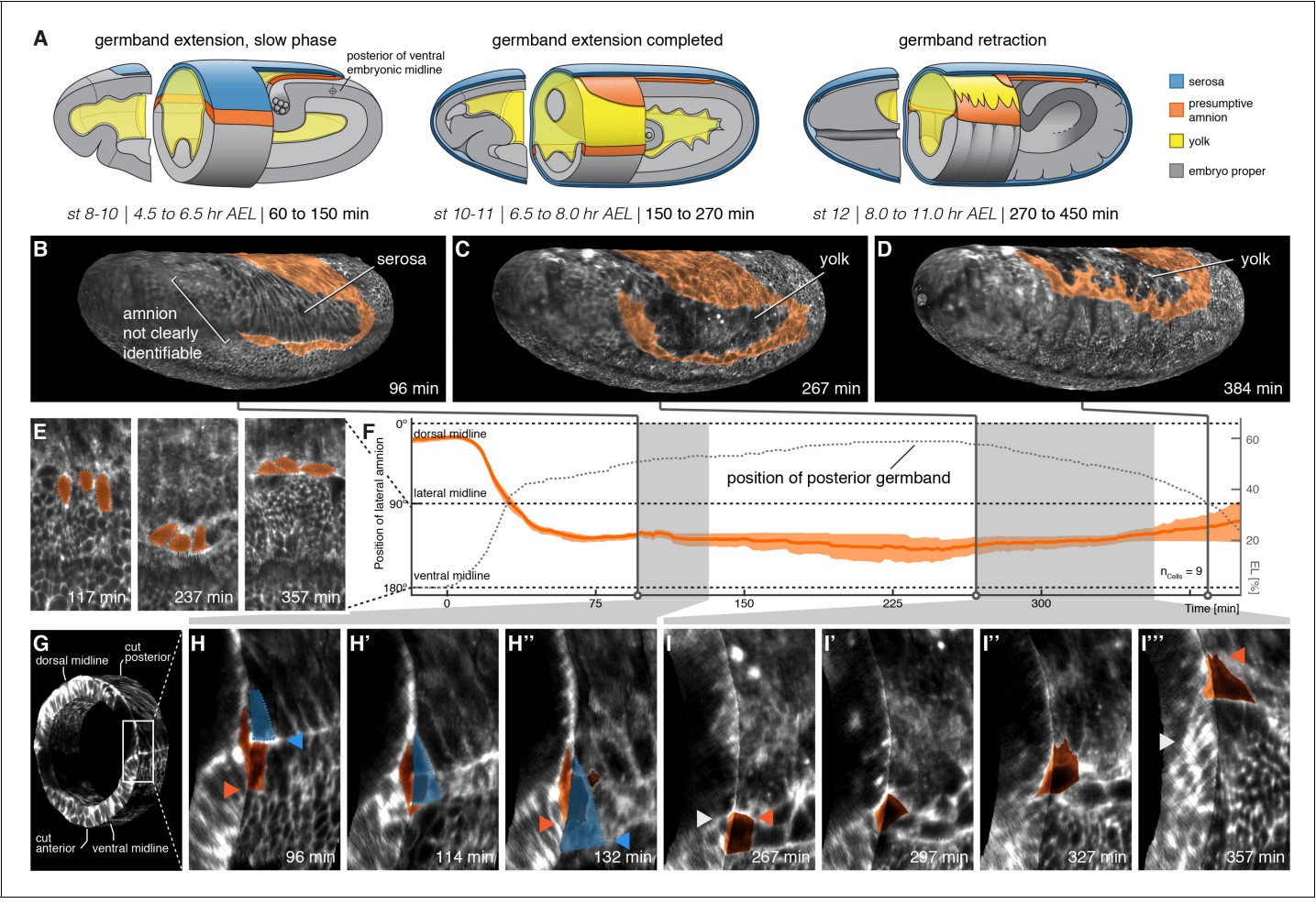

**Figure 2.** The lateral anlage of the *M. abdita* putative amnion differentiates into a one-to-two cell wide lateral epithelium. (A) Model of extraembryonic tissue development in *M. abdita* based on SPIM time-lapse recordings; staging as in *Figure 1*. (B–D) Global embryonic views of SPIM recorded embryos at corresponding stages. To reveal and mark the putative amnion, the surface layer has been digitally removed (*Figure 2—figure supplement 1*). (E,F) Tracking (E) and plotting (F) of amnion cell position (orange) along the dorso-ventral circumference (0° corresponds to dorsal, 180° to ventral midline) relative to position of extending germband as measure of developmental progression (dotted line, 0% egg length (EL) at the posterior pole), standard error of mean shown as shades. (G–I''') Donut section of SPIM recorded image volume (G) to illustrate close-up views of amnion cell behavior as the serosa (blue) detaches during germband extension (H) and after onset of germband retraction (I). As the serosa spreads over the amnion (H–H''), the amnion appears to bend underneath the serosa. The serosa then separates from the amnion and spreads over the embryo proper. During germband retraction (I–I'''), the amnion starts to extend actin-rich protrusions and leads the ectoderm as the tissue progresses towards the dorsal midline. Triangles indicate relative positions of amnion, serosa, and ectoderm in first and last time points (amnion = orange, serosa = blue, and embryonic cells = grey). Embryos are shown with anterior left and dorsal up.

DOI: https://doi.org/10.7554/eLife.34616.004

The following video and figure supplement are available for figure 2:

**Figure supplement 1.** Computational removal of embryo surface layers was used to reveal and mark putative amnion cells underneath the serosa.
DOI: https://doi.org/10.7554/eLife.34616.005

**Figure 2—video 1.** The video corresponds to panels H-H'' of *Figure 2*.
DOI: https://doi.org/10.7554/eLife.34616.006

**Figure 2—video 2.** The video corresponds to panels I-I''' of *Figure 2*.
DOI: https://doi.org/10.7554/eLife.34616.007

F-actin (revealed by our Lifeact reporter) between the two cell types as local hallmark. Following this hallmark, amnion cells initially remained in contact with the F-actin front and appeared to get dragged over the adjacent ectoderm by the steadily expanding serosa (*Figure 2H,H'*). After having been passed over by the serosa, amnion cells stopped moving towards the ventral midline. By

contrast, the F-actin front continued to expand evenly towards the ventral midline together with the leading edge of the serosa (*Figure 2H''*), indicating that serosa leading edge and lateral amnion had disjoined. Following this disjunction, the presumptive amnion cells seemed to be turned with their lateral side towards the basal membrane of the serosa (*Figure 2B,H*; *Figure 2—video1*). This orientation was reversed during germband retraction, when amnion cells no longer appeared folded over the adjacent ectoderm but rather over the yolk sac (*Figure 2D,I–I'''*; *Figure 2—video2*).

## The serosa in *M. abdita* expands in distinct phases

Previous analyses of *M. abdita* serosa expansion have been based on the expression of specific marker genes in fixed embryos (*Rafiqi et al., 2008*) as well as on bright field microscopy time-lapse recordings (*Wotton et al., 2014*). These analyses suggest that the serosa starts to be detectable about 30–45 min after the onset of germband extension; it then expands, presumably homogeneously, for almost 2.5 hr until the tissue front passes the equator at both poles of the embryo; and then, within minutes, it fuses rapidly along the ventral midline (*Wotton et al., 2014*).

We revisited these dynamics in our SPIM recordings by marking the serosa in projected carpets, similar as outlined above for the amnion. Marking of the serosa was guided by cell size as well as the enrichment of the actin reporter at the interface of amnion and serosa (*Figure 2H*, *Figure 3—figure supplement 1*), which suggested the formation of a supracellular actin cable at the serosa boundary similar to the actin cable observed during serosa formation in the flour beetle *Tribolium castaneum* (*Benton et al., 2013*). Accordingly, we identified first signs of tissue expansion at about 35 min after onset of germband extension (*Figure 3A*; *Figure 3—video1*; *Figure 3—video2*). The serosa approached the equator at the poles at about 150 min (*Figure 3B*) and passed it 15 min later; ventral fusion along the midline was then completed at about 240 min (*Figure 3C*). Our measures of serosa area increase were overall in accordance with previous analyses (*Figure 3D,E*). The increased resolution in ventral views allowed us to determine the specific dynamics of ventral fusion and suggested a closure rate of about 11 µm/min along the anterior-to-posterior axis of the serosal window, and 3 µm/min across (*Figure 3F,G*), which was considerably faster than rates that have been described for dorsal closure of the amnioserosa in *D. melanogaster* (*Hutson et al., 2003*). Notably, our continuous mapping of area expansion indicated that *M. abdita* serosa spreading was a non-homogeneous process, in which two periods of almost linear area growth were interrupted by a roughly 50-min interval without substantial tissue area increase (70–120 min, *Figure 3D*). At the end of this interval, we observed disjunction of the serosa from adjacent tissue: first at the anterior, then the posterior and finally at its lateral sides (*Figure 3E*).

## Decoupling of serosa and yolk sac is preceding serosa tissue spreading

The observed interruption in serosa area increase hinted towards a possible mechanism of interfering with tissue spreading. For example, the interruption in serosa expansion could be explained by a temporal pause in a tissue-autonomous program of cell thinning and spreading. However, gene expression of the tissue-fate determining Hox-3 transcription factor Zen is reportedly high also throughout stages in which tissue expansion paused (*Rafiqi et al., 2008*), indicating little, if any, change in potential upstream genetic regulation of cell thinning and spreading. Alternatively, serosa area increase could be paused if further tissue spreading was hindered by adhesion to an underlying substrate. A possible candidate for such a substrate is the yolk sac (*Schmidt-Ott and Kwan, 2016*).

To test whether interaction between yolk sac and serosa could explain the pause in tissue spreading, we aimed to test for mechanical interaction between yolk sac and serosa. To generate a reporter that could visualize the yolk sac, we first cloned the *M. abdita* orthologue of *basigin* (*Mab-bsg*, *Figure 4—figure supplement 1*), which encodes a trans-membrane protein that in *D. melanogaster* is enriched in the yolk sac membrane (*Reed et al., 2004*; *Goodwin et al., 2016*). Analysis of *Mab-bsg* expression suggested that the gene was expressed in the yolk sac (*Figure 4—figure supplement 1*). We then asked whether a fusion of Basigin and eGFP (Basigin-eGFP) could be used to visualize the yolk sac membrane like in *D. melanogaster* (*Goodwin et al., 2016*). To analyze expression of Basigin-eGFP together with ubiquitously marked membranes of the serosa, we first injected Lifeact-mCherry into early syncytial embryos. Then, to express the Basigin reporter specifically in the yolk sac, we injected capped mRNA encoding the fusion protein after germband extension had started and cellularization was presumably complete (*Rafiqi et al., 2010*). Visual inspection

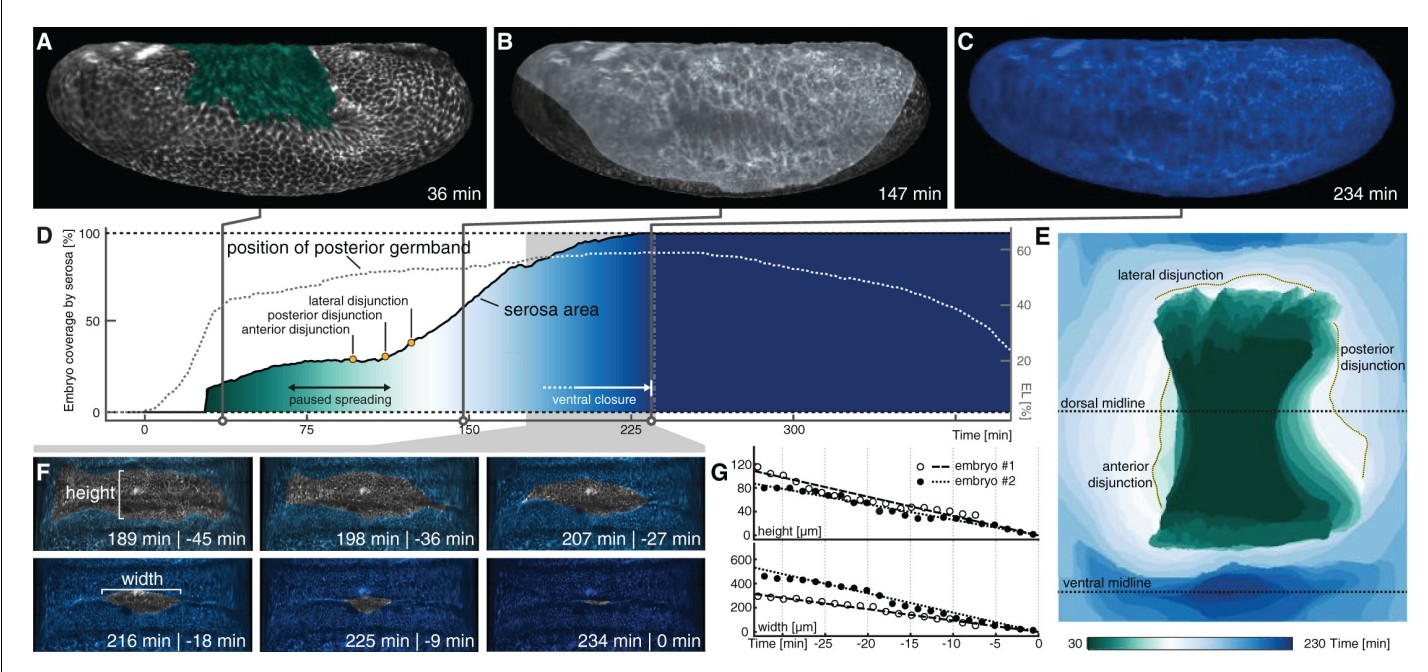

**Figure 3.** Serosa expansion in *M. abdita* is characterized by two distinct phases that are separated by a pause in tissue spreading. (**A–C**) Onset (**A**), passing of poles (**B**), and ventral closure (**C**) of marked serosa illustrate previously described stages of expansion (***Wotton et al., 2014***). (**D,E**) Plotting of serosa area over time and relative to position of extending germband (**D**) indicates two additional stages of serosa expansion, that is a pause in spreading and the subsequent disjunction, first at the anterior, then posterior, and finally lateral circumference (**E**). Time is indicated in colormap. (**F,G**) Progression of ventral closure indicated in projections of ventral embryo surfaces (0 = ventral closure) (**F**) and quantified as linear decrease in height and width of serosal window (**G**).

DOI: https://doi.org/10.7554/eLife.34616.008

The following video and figure supplement are available for figure 3:

**Figure supplement 1.** Cell tracks, cell shape, and enrichment of the Lifeact reporter at the serosa periphery were used as landmarks to mark the amnion.

DOI: https://doi.org/10.7554/eLife.34616.009

**Figure 3—video 1.** The video corresponds to panels A-C of *Figure 3*.

DOI: https://doi.org/10.7554/eLife.34616.010

**Figure 3—video 2.** The video corresponds to a ventral view of panels A-C of *Figure 3*.

DOI: https://doi.org/10.7554/eLife.34616.011

of time-lapse recordings along the dorsal midline indicated that fluorescent signals in serosa and yolk sac could be separated, allowing us to distinguish between movements in either tissue (*Figure 4A–C*; *Figure 4—video1*).

With Basigin-eGFP established as reporter for the yolk sac membrane in *M. abdita*, we used its fluorescent signal to track movements at the yolk sac surface by optical flow (see Materials and methods). These analyses detected potential oscillations in membrane behavior along the anterior-to-posterior axis, which seemed to coincide with similar behavior in individual serosa cells (*Figure 4D*). The direct comparison of general movements in yolk sac and serosa revealed a strong positive correlation, which was very specific to individual serosa cells and the yolk sac membrane directly underneath (*Figure 4E,F*, and *Figure 4—figure supplement 1*). Such positive correlation of movements is indicative of strong mechanical coupling between yolk sac and extraembryonic tissue (*Goodwin et al., 2016*). We found coupling to be significantly reduced after the second phase of serosa expansion had started (*Figure 4E,F*). Notably, coupling was not completely lost, suggesting that yolk sac and serosa remained physically associated, but loosely enough to slide past each other. Taken together, our results indicated that free spreading of the *M. abdita* serosa over the embryo was preceded by a decoupling of serosa and underlying yolk sac during paused tissue spreading (*Figure 4—figure supplement 2*).

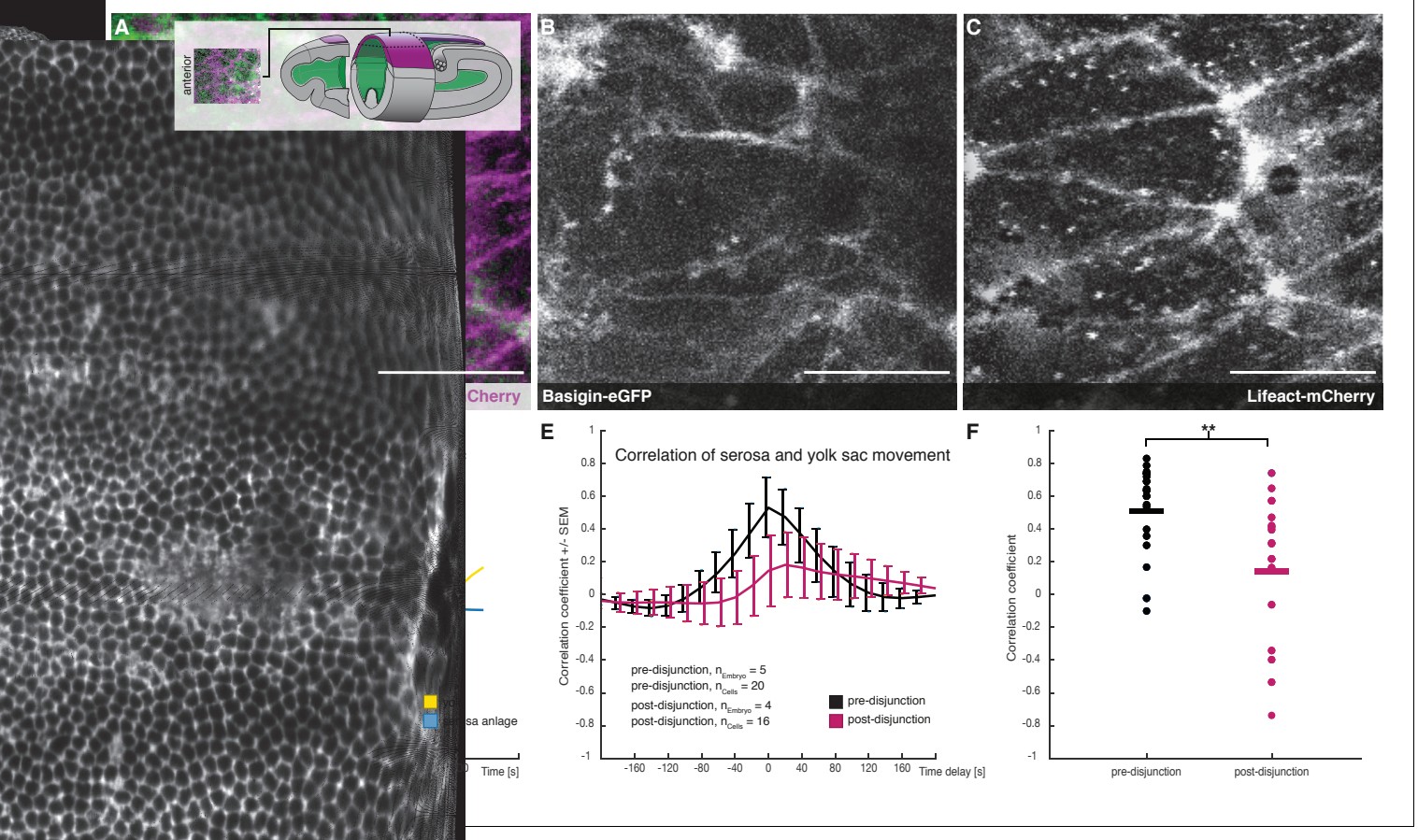

**Figure 4.** Cross-correlation of serosa cells and yolk sac movements suggest a decoupling of serosa prior to free spreading. (A–C) Representative sample images of serosa (visualized with Lifeact-mCherry) and underlying yolk sac (expressing Basigin-eGFP) are shown in average-intensity projections. Images are stills from time lapse-recordings taken along the dorsal midline and anterior of the extending germband, where serosa and yolk sac were in direct contact (schematic illustration provided in inset). (D) Adjusted serosa (blue) and yolk sac (yellow) displacement measured by optical flow analysis exemplary for one serosa cell and corresponding substrate area underneath. (E) Average cross-correlation function of cells and substrate movements indicate coupling before (black) and decoupling after (magenta) serosa disjunction of the serosa from the embryo proper. Standard error of mean shown as bars. (F) Collective comparison of correlation coefficients for individual cells before and after serosa disjunction, bar indicates the mean. **p=0.00686 based on Student's t-test. Anterior is to the left. Scale bars, 20 μm.

DOI: https://doi.org/10.7554/eLife.34616.012

The following video and figure supplements are available for figure 4:

**Figure supplement 1.** Expression of *M. abdita basigin* (*Mab-bsg*) and controls for cross correlation analysis in wildtype embryos.

DOI: https://doi.org/10.7554/eLife.34616.013

**Figure supplement 2.** Timeline to outline reported features of extraembryonic development in the context of *M. abdita* embryonic staging.

DOI: https://doi.org/10.7554/eLife.34616.014

**Figure 4—video 1.** The video corresponds to panel A and quantifications presented in panels E and F of *Figure 4*.

DOI: https://doi.org/10.7554/eLife.34616.015

## *Mab-Mmp1* modulates tissue–tissue interaction between yolk sac and serosa

In *D. melanogaster*, modulation of tissue-tissue adhesion as well as complete tissue-tissue decoupling has been previously associated with *Matrix metalloprotease 1* (*Mmp1*) activity (*Diaz-de-la-Loza et al., 2018*; *Glasheen et al., 2010*; *LaFever et al., 2017*; *Srivastava et al., 2007*). To test whether activity of this matrix metalloprotease was involved in mechanical decoupling of serosa and yolk sac, we cloned the *M. abdita* orthologue of *Mmp1* (*Mab-Mmp1*) and found it expressed in yolk sac nuclei (*Figure 5A*). To test whether *Mab-Mmp1* had an effect on interaction between serosa and

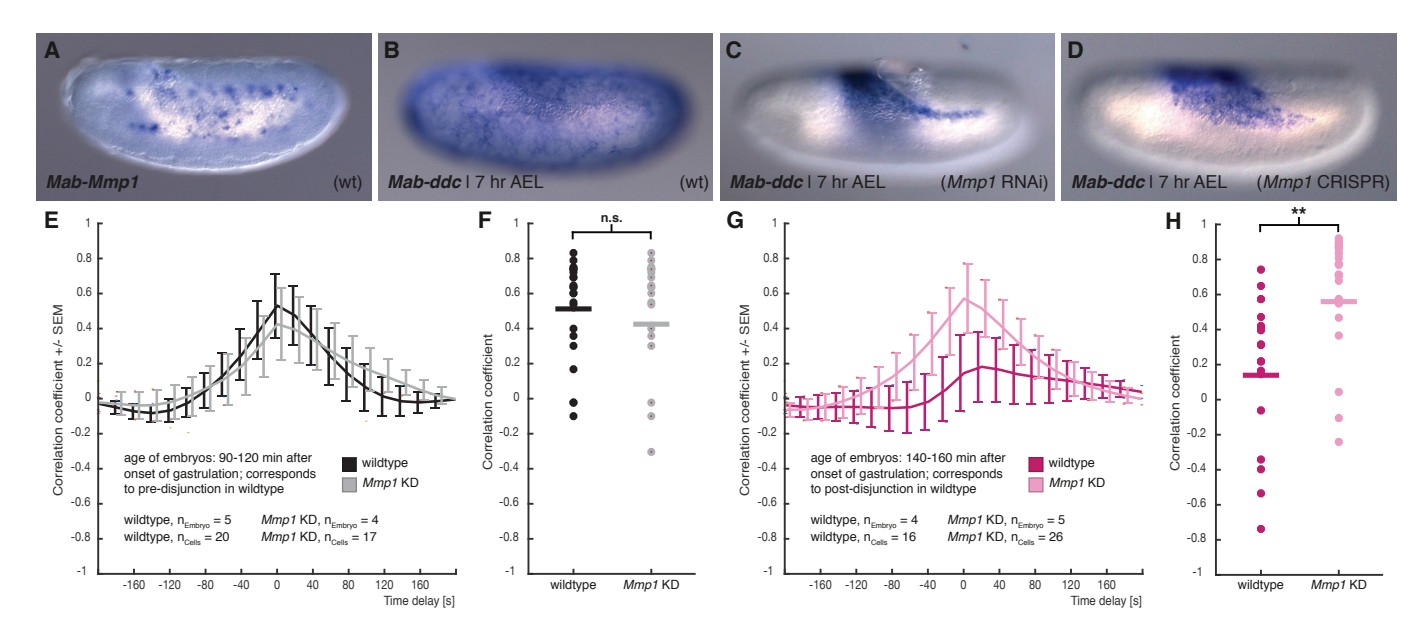

Figure 5. *M. abdita Matrix metalloprotease 1* (*Mab-Mmp1*) regulates serosa decoupling. (**A**) Expression of *Mab-Mmp1* during germband extension stage. (**B–D**) Expression of *M. abdita dopa decarboxylase* (*Mab-ddc*) as serosa marker during germband extension stage in wildtype (**B**), in *Mab-Mmp1* RNAi (**C**), and *Mab-Mmp1* CRISPR/Cas9 embryos (**D**). (**E,F**) Average cross-correlation function of corresponding cell and substrate movements indicate similar level of mechanical coupling in wildtype (black) and *Mab-Mmp1* RNAi embryos (grey) at the end of serosa expansion pause in wildtype embryos (**E**), with statistical support from correlation coefficients of individual cells (**F**). (**G,H**) Average cross-correlation function of corresponding cells and substrate movements indicate low mechanical coupling in wildtype (dark magenta) and high mechanical coupling in *Mab-Mmp1* RNAi embryos (light magenta) after onset of free serosa expansion in wildtype embryos (**G**), with statistical support from correlation coefficients for individual cells (**H**). Bars indicate mean. **p=0.00279; n.s. p=0.48746, based on Student's *t*-test.

DOI: https://doi.org/10.7554/eLife.34616.016

The following video and figure supplement are available for figure 5:

**Figure supplement 1.** Expression of *Mab-ddc* as serosa marker and controls for cross correlation analyses in *Mab-Mmp1* RNAi embryos.
DOI: https://doi.org/10.7554/eLife.34616.017

**Figure 5—video 1.** The video corresponds to quantifications presented in Panels E-H in *Figure 5*.
DOI: https://doi.org/10.7554/eLife.34616.018

yolk sac, we analyzed embryos in which gene activity was knocked down using RNAi or CRISPR. We first tested for putative effects on serosa development by staining embryos for the serosa-specific marker gene *dopa decarboxylase* (*Mab-ddc*; (*Rafiqi et al., 2010*)). In wildtype embryos, the serosa closes about 7 to 7.5 hr after egg lay (*Figure 3D*), which was reflected in uniform staining of *Mab-ddc* in correspondingly staged and fixed embryos (*Figure 5B*, (*Rafiqi et al., 2010*)). Following knockdown by RNAi (96%, n = 100/104) and CRISPR (56%, n = 77/136), *Mab-ddc* expression was reduced at corresponding stages to a dorsal domain, suggesting that serosa spreading was impaired (*Figure 5C,D*).

To address how knockdown of *Mab-Mmp1* affected mechanical coupling of serosa and yolk sac, we quantified tissue-level movements and correlated flow velocity in RNAi embryos before and after presumptive decoupling. In *Mab-Mmp1* RNAi embryos corresponding to wildtype developmental stages prior to decoupling (90 – 120 min), the correlation of movements in serosa and yolk sac were comparable to wildtype and suggested mechanical coupling between the two membranes (*Figure 5E,F*; *Figure 5—video1*). In *Mab-Mmp1* RNAi embryos corresponding to wildtype developmental stages after decoupling (140 – 200 min), correlation of serosa and yolk sac movements remained virtually unchanged and were significantly higher than in wildtype (*Figure 5G,H*; *Figure 5—video1*). These results indicated that mechanical coupling between the two membranes was not broken up, thus explaining impaired serosa spreading.

To determine whether serosa spreading was attenuated, delayed, or completely halted after knockdown of *Mab-Mmp1*, we analyzed serosa spreading by marker gene expression in older embryos, that is 9 hr and 11 hr AEL (*Figure 5—figure supplement 1*). In 9-hr old *Mab-Mmp1* RNAi embryos, the serosa was not yet fused; in many embryos it expanded to the mid-lateral side (33%, n = 27/83), and a few embryos had strong phenotype with a dorsal serosa (4%, n = 3/83). In 11-hr old *Mab-Mmp1* RNAi embryos, the serosa was fused in most embryos (77%, n = 30/39), while a few specimen still showed the non-fused phenotype (23%, n = 9/39). These results suggested that serosa spreading was attenuated or delayed after knockdown of *Mab-Mmp1*. To explain the observed delay in the completion of serosa closure, we cannot exclude a possibly incomplete knockdown of *Mab-Mmp1* activity. However, we consider it more likely that *Mab-Mmp1* acts redundantly with yet unknown factors to release the serosa from its yolk sac attachment.

## Discussion

With our work we established the scuttle fly *M. abdita* as a functionally accessible model to dissect cellular dynamics of extraembryonic tissue spreading in flies. Imaging the early *M. abdita* embryo over an extended period of time has allowed us to place the dynamics of extraembryonic development into the context of the developing embryo, at a spatiotemporal resolution that is comparable to previous work in *D. melanogaster* and the flour beetle *T. castaneum* (*Benton et al., 2013*; *Chhetri et al., 2015*; *Goodwin et al., 2016*; *Hilbrant et al., 2016*). This made it possible to connect previously studied pattern formation and signaling in the blastoderm with progression of cell and tissue differentiation (*Kwan et al., 2016*; *Rafiqi et al., 2008*; *Rafiqi et al., 2012*). The combination of long-term and *in toto* imaging enabled us to address open questions regarding amnion and serosa formation in *M. abdita*, and it allowed us to identify tissue-tissue interactions between yolk sac and serosa as a mechanism that controls extraembryonic tissue spreading in *M. abdita*.

### *M. abdita* forms an open lateral amnion

Our cell and tissue tracking in SPIM recordings of early *M. abdita* development were consistent with previously reported expression of a late amnion marker gene, M. abdita eiger (*Mab-egr*; (*Kwan et al., 2016*)). Our time-lapse recordings and previously published *Mab-egr* expression indicated that the *M. abdita* amnion developed as an open and mostly lateral tissue. This observation correlates with the size of its blastoderm anlage, which is dorsolaterally narrow and slightly larger only in the posterior blastoderm (*Kwan et al., 2016*; *Rafiqi et al., 2012*). Tissue topology of the amnion in *M. abdita* is thus markedly different than the closed and ventral amnion that has been reported for non-cyclorrhaphan flies and most other insects (*Panfilio, 2008*; *Schmidt-Ott and Kwan, 2016*). For a possible explanation of this peculiar topology we revisited the idea that the amnion in *M. abdita* has undergone different degrees of reduction (*Schmidt-Ott et al., 2010*). While the posterior amnion anlage then is likely to reflect a more ancestral size and thus is still sufficient to form a vestigial ventral amnion, the lateral amnion is secondarily reduced (*Kwan et al., 2016*; *Rafiqi et al., 2012*), and the resulting tissue insufficient in area to close over the ventral side of the embryo proper. The functional implications of this reduction on embryonic fly development are difficult to assess, in part because the precise function of the ventral amnion in insects is still unknown (*Panfilio, 2008*). Further work is required to elucidate genetic mechanisms that distinguish a lateral and a ventral amnion, which could develop from a larger anlage, rely on proliferating or highly stretchable amnion cells, or, like in *T. castaneum*, benefit from a germband that participates in ventral amniotic cavity formation (*Benton, 2018*).

### *M. abdita* serosa spreading proceeds in distinct phases

Area tracking of *M. abdita* serosa development identified two distinct phases of tissue spreading. These two phases corresponded to the early 'tethered' and the late 'freed' state of the serosa and were interrupted by a notable pause in tissue area expansion. In the first phase of serosa spreading, the serosa was still continuous with amnion and ectoderm and thus part of a coherent epithelium. During this phase, the initially columnar blastoderm cells changed their shape and became squamous. In the second phase, the serosa was characterized by free edges at its periphery, cells increased further in their apical area, the tissue spread over the entire embryo proper, and eventually fused along the ventral midline. The dynamics of this process have been previously difficult to

observe, presumably because the ventral serosa is difficult to discern in DIC images. Here we report that ventral closure of the serosa in *M. abdita* was substantially faster than dorsal closure of the *D. melanogaster* amnioserosa, and we propose that the comparatively fast fusion rate in the serosa can be explained by lack of segmental alignment and time consuming zippering as observed for dorsal closure in *D. melanogaster* (*Jacinto et al., 2002*).

## Free serosa spreading in *M. abdita* requires decoupling from yolk sac

Closer analysis of the transition from paused to free serosa spreading provided evidence for a change in tissue–tissue interaction between serosa and the underlying yolk sac. Strong correlation of movements in serosa and yolk sac membranes indicated a tight coupling between the membranes in the phase of paused serosa spreading, which could be observed until the extending germband reached about the middle of the egg. After that, correlation of movements between serosa and yolk sac was significantly reduced and disjunction of serosa and amnion could be observed. Notably, serosa disjunction was a gradual process that was first observed in the anterior and last in its lateral periphery. Regardless of location and timing, disjunction was characterized by a very steady and even expansion of the serosa leading edge without signs of tissue rupture, suggesting that tissue-level decoupling from the yolk sac and cell-level loss of adhesion from the amnion had already occurred earlier. We found that coupling of serosa and yolk sac persisted in *Mab-Mmp1* RNAi embryos substantially longer, and interactions between the two tissues remained comparable to that of wildtype embryos during paused serosa spreading. Complementing these findings, we found serosa spreading impaired in *Mab-Mmp1* RNAi embryos. Taken together, our results strongly suggest that free serosa spreading in *M. abdita* requires its decoupling from the yolk sac.

Similar interactions between yolk sac and extraembryonic tissue have been reported previously in *D. melanogaster*, albeit at later stages of development, where they contribute to germband retraction and dorsal closure (*Goodwin et al., 2016*; *Narasimha and Brown, 2004*; *Reed et al., 2004*; *Schöck and Perrimon, 2003*). More generally, interactions of yolk sac and overlying epithelia have been long implicated in insect as well as vertebrate development (*Anderson, 1972a*; *Anderson, 1972b*; *Bruce, 2016*; *Counce, 1961*; *Schmidt-Ott and Kwan, 2016*), suggesting that yolk sac-dependent regulation of serosa spreading in the *M. abdita* embryo may reflect a more common phenomenon.

## On the origin of the amnioserosa

Genetic changes that affect either serosa differentiation or the contact between serosa and yolk sac in *M. abdita* could have similarly occurred in the last common ancestor of *D. melanogaster* and flies with separate amnion and serosa. To explain the origin of the amnioserosa by such ancient genetic changes, the current model focuses on serosa differentiation by *zen* activity. It stresses the absence of postgastrular *zen* activity in *D. melanogaster*, outlines how terminal serosa differentiation is impaired as a consequence, and suggests that the resulting de-repression of amnion development has led to the formation of a non-spreading dorsal extraembryonic tissue (*Rafiqi et al., 2008*; *Rafiqi et al., 2010*; *Schmidt-Ott and Kwan, 2016*). Motivated by our results, and in the absence of a functionally described mechanism of postgastrular *zen* repression, we considered the possibility that the amnioserosa may have alternatively originated by non-resolved tissue coupling between yolk sac and serosa. Topologically, and presumably also functionally, the resulting composite tissue of a non-spreading serosa and peripheral amnion would have shared similarities with the amnioserosa of *D. melanogaster*. Complementing biomechanical and gene expression analyses in the amnioserosa support the idea that the amnioserosa has been, or still is, a composite tissue (*Gorfinkiel et al., 2009*; *Wada et al., 2007*), and *Mmp1* is not expressed in the yolk sac of *D. melanogaster* (*Page-McCaw et al., 2003*). To account for complete loss of serosa decoupling and the expression of amnion genes in the *D. melanogaster* amnioserosa (*Rafiqi et al., 2008*; *Rafiqi et al., 2010*), additional secondary mutations would need to be postulated. Such conditions provided, however, the amnioserosa could have originated in the presence of postgastrular *zen* activity through ancient changes in tissue-tissue interactions.

## Materials and methods

### Fly stocks

The laboratory culture of *Megaselia abdita* (Sander strain) was maintained at 25°C and a constant 16/8 hr day/night cycle as described previously (*Caroti et al., 2015*).

### Cloning and RNA synthesis

*Mab-Mmp1* was identified from genome and transcriptome sequences. A genomic fragment (MK090468) was cloned after PCR amplification from the locus, a cDNA fragment after amplification through 5'-RACE (MK090469). Double-stranded RNA (dsRNA) was synthesized as described (*Urbansky et al., 2016*); *Mab-Mmp1* dsRNA comprised pos. +103 to +1167 of the genomic fragment (pos. one refers to first nucleotide in ORF) and included a 57 bp intron at pos. +575. Guide RNAs for a knock-out of *Mab-Mmp1* were designed using CCTop as CRISPR/Cas9 target online predictor (*Stemmer et al., 2015*). Three single guide RNAs (sgRNAs) were designed to target the following positions (pos. one refers to first nucleotide in ORF): sgRNA1, 5'-TGCAGAGCGTATCTCTTT, pos +404 to +387; sgRNA2, 5'-CGTGGACTATTGATTGTC, pos +710 to +693; sgRNA3, 5'-TCGGCAACCGAGTTTTCA, pos +898 to+881. Guide RNAs as well as Cas9 mRNA were synthesized as described (*Stemmer et al., 2015*).

*Mab-bsg* was identified from genome and transcriptome sequences. A fragment encompassing the full open reading frame was PCR amplified and used in a Gibson Assembly to generate a 3' fusion with eGFP in a pSP expression vector (pSP-*Mab-bsg-eGFP*). RNA was *in vitro* transcribed using SP6 Polymerase (Roche), capping and polyA-tailing was performed using ScriptCap Cap 1 Capping System and Poly(A) Polymerase Tailing Kit (CellScript).

### Preparation of Lifeact-eGFP, Lifeact-mCherry, and Histone H1

Heterologous expression vectors for recombinant Lifeact-eGFP and Lifeact-mCherry were generated by cloning PCR-amplified constructs into pET-21a(+). The fragment encoding for Lifeact-eGFP was amplified from pT7-LifeAct-EGFP (*Benton et al., 2013*) using primer pair 5'-AAACATATGGGCGTGGCCGATCTGAT/5'-TTTTCTCGAGCTTGTACAGCTCGTCCATGC, digested with *Nde*I and *Xho*I, and cloned into pET-21a(+) to generate pET-Lifeact-eGFP. Similarly, a fragment encoding for mCherry was amplified from H2Av-mCherry (*Krzic et al., 2012*) using primer pair 5'-GAGGGGATCCTCGCCACCAGATCCATGGTGAGCAAGGGCGAGGAG/5'-GGTGCTCGAGGGCGCCGGTGGAGTGGCGGCC, digested with *Bam*HI and *Xho*I, and replaced the eGFP-encoding fragment in pET-Lifeact-eGFP with mCherry to generate pET-Lifeact-mCherry.

Recombinant Lifeact-FP protein was expressed in *E.coli* BL21 after induction with IPTG (final concentration 1 mM) at $OD_{600} = 0.6 - 0.8$. Cells were pelleted 4 hr after induction, washed in PBS, and resuspended in lysis buffer on ice (50 mM NaPO4 pH 8.0, 0.5 M NaCl, 0.5% glycerol, 0.5% Tween-20, 10 mM imidazole, 1 mg/ml lysozyme). Resuspended cells were sonicated with 15 – 30 s pulses, centrifuged, and the supernatant mixed with equilibrated Ni-NTA agarose beads (Cube Biotech, Germany). Protein binding was carried out for 2 hr at 4°C, beads were washed three times at high-salt/high-pH (50 mM NaPO4 pH 8.0, 250 mM NaCl, 0.05% Tween-20, 20 mM Imidazole), once at high-salt/low-pH (50 mM NaPO4 pH 6.0, 250 mM NaCl, 0.05% Tween-20, 20 mM Imidazole), and twice at high-salt/high-pH without detergent (50 mM NaPO4 pH 8.0, 250 mM NaCl, 20 mM Imidazole). Following the washes, beads were transferred into a poly-prep chromatography column (Bio-Rad Laboratories) and the protein was eluted in multiple aliquots of elution buffer (50 mM NaPO4 pH 8.0, 150 mM NaCl, 250 mM Imidazole, 5% Glycerol). Collected protein fractions were analyzed by SDS-PAGE and dialyzed against PBS. Final concentrations were typically around 0.5 mg/ml; aliquots were stored at −80°C.

Histone H1 (Merck/Calbiochem) was fluorescently tagged using Texas Red-X Protein Labeling Kit (ThermoFisher) as described (*Mori et al., 2011*). Final concentration was typically around 2 mg/ml; 10% saturated sucrose was added as anti-frost reagent, and aliquots were stored at −80°C.

### Immunohistochemistry

For whole mount in situ hybridization using NBT/BCIP as stain, embryos were heat fixed at indicated time points after developing at 25°C, devitellinized using a 1 + 1 mix of n-heptane and methanol,

and post-fixed using 5% formaldehyde as described (*Rafiqi et al., 2011a*). For staining with Phallacidin and DAPI, embryos were fixed with 4% formaldehyde and devitellinized using a 1 + 1 mix of n-heptane and 90% ethanol. Whenever necessary, manual devitellinization was performed as described (*Rafiqi et al., 2011a*).

RNA probe synthesis, whole mount in situ hybridization, and detection was carried out as described (*Lemke and Schmidt-Ott, 2009*). The following cDNA fragments were used as probes: *Mab-ddc* (*Rafiqi et al., 2010*), *Mab-egr* (*Kwan et al., 2016*), and the newly cloned *Mab-Mmp1* and *Mab-bsg* cDNA fragments. Phallacidin staining was performed as described (*Panfilio and Roth, 2010*) with modifications: the stock (200 units/ml, Invitrogen B607) was diluted in PBS (1:25), embryos were stained for 3 hr at room temperature and then briefly rinsed three times in PBS. DNA was stained using 4',6-diamidino-2-phenylindole (DAPI, Life Technology D1306) at a final concentration of 0.2 µg/ml.

## Injections

Embryos were collected, prepared for injection, and injected essentially as described (*Rafiqi et al., 2011b*). dsRNA was injected with concentrations of 3.9 mg/ml, which corresponded to about 6 µM of *Mab-Mmp1* dsRNA. Concentration of injected *Mab-bsg* mRNA was about 3.3 mg/ml; concentration of injected Lifeact-mCherry protein was about 0.5 mg/ml, Histone H1 was injected at concentrations of about 0.7 mg/ml. Cas9 mRNA and all three sgRNAs were co-injected as a mix with a final concentration of 1 mg/ml of Cas9 mRNA and 50 ng/ml for each of the sgRNAs (*Bassett et al., 2013*).

## Microscopy

Histochemical staining was recorded with DIC on a Zeiss Axio Imager M1 using 10x (dry, 10x/0.45); embryos were embedded in a 3 + 1 mix of glycerol and PBS. In the absence of temperature-controlled stages, all time-lapse recordings have been acquired at room temperature. For both, confocal tissue recordings and initial *in toto* bright-field reference observations, room temperature was frequently monitored at 25°C.

## Confocal live imaging

Embryos were injected in the syncytial blastoderm stage with either recombinant Lifeact-mCherry (wildtype analyses) or a 1:1 mix of Lifeact-mCherry and *Mab-Mmp1* dsRNA. Because a protein trap line equivalent to the *D. melanogaster basigin* reporter does not exist in *M. abdita* (*Reed et al., 2004*), we reasoned that injection of mRNA encoding *Mab-bsg-eGFP* into the yolk sac could mimic the desired properties of the *D. melanogaster* reporter as closely as possible and with minimal side effects in *M. abdita*. To express *Mab-bsg-eGFP* specifically in the yolk sac, capped mRNA of the reporter was injected after germband extension had started and yolk sac formation is thought to be completed (*Rafiqi et al., 2010*). Time lapse recordings were taken along the dorsal midline anterior to the extending germband where serosa and yolk sac were in contact. For the analysis of early, pre-disjunction tissue interaction, recordings were taken 90 – 120 min after onset of germband extension; for the analysis of a post-disjunction tissue interaction, recordings were taken 140 – 190 min after onset of germband extension. Recordings were made by single-photon confocal imaging on a Leica system (SP8) using a 63x immersion objective (HC PL APO 63x/1.30 Glyc CORR CS2). Volumes were recorded in 15 to 20 confocal sections of 1 µm with simultaneous detection of mCherry and eGFP. Voxel size was 0.24 × 0.24×1 µm and volumes were collected at 20-s intervals for 10 min.

## Light-sheet microscope setup and imaging

Time-lapse recordings were performed using two Multiview light-sheet microscopes (MuVi-SPIM) (*Krzic et al., 2012*) with confocal line detection (*de Medeiros et al., 2015*). The microscopes were equipped with two 25 × 1.1 NA water immersion objective lenses (CFI75 Apo LWD 25XW, Nikon) or two 16 × 0.8 NA water immersion objective lenses (CFI75 Achro LWD 16XW, Nikon) for detection. Illumination was performed via two 10 × 0.3 NA water immersion objective lenses (CFI Plan Fluor 10XW). All objectives were used with the corresponding 200 mm tube lenses from Nikon. Fluorescence of mCherry was excited at 561 nm or 594 nm, TexasRed at 642 nm. Fluorescence was imaged simultaneously onto two sCMOS cameras (Hamamatsu Flash 4 V2) after passing corresponding

fluorescence filters on the detection paths (561 nm LP, 647 nm LP, 594 nm LP, EdgeBasic product line, Semrock).

*M. abdita* embryos were injected in oil (refractive index 1.335, Cargille Labs) and mounted in an oil-filled fluorinate ethylene propylene (FEP) tube (*Kaufmann et al., 2012*). This tube was stabilized with a glass capillary that was placed into the capillary holder of the microscope. All embryos were imaged from four sides (one 90° rotation) every 1.5 or 2 min with a z-spacing of 1 μm for membrane labeled embryos and 2 μm for nuclear labeled embryos. The four orthogonal views facilitated a more uniform sampling, and the typical exposure time per plane of around 40 ms guaranteed an overall high temporal resolution. The resultant four stacks per time point were fused using previously published software (*Krzic et al., 2012*; *Preibisch et al., 2010*). All further processing and analysis was performed on the fused data sets. Analysis of *M. abdita* embryonic development was based on a total of 3 wildtype MuVi-SPIM recordings.

## Generation of embryo point clouds at and below the surface level

To allow for rapid image operations, fused 3D image stacks of individual MuVi-SPIM time points were transformed into point clouds. For this, fused 3D image stacks were read into Matlab using *StackReader*. Time-adaptive intensity thresholding was then used to segment the 3D image stacks into exactly two solid components: embryo and background. If segmentation returned more than one object, all but the largest one were eliminated and holes resulting from a lower fluorescence intensity in the yolk area were filled (*Segmentation*). To reveal fluorescent signal in layers below the embryo surface, the outermost layer of the segmented embryo was eroded using morphological operators (*imerode*) and a kernel radius of the specified depth. When needed, the surface was smoothened through morphological closing or opening (*imopen*, *imclose*). To visualize fluorescent signal for a specific layer of the embryo, the embryo was eroded at different depths and the resulting images subtracted from the original producing a set of concentric layers (*OnionCheat*). Point clouds were generated by mapping the geometrical voxel information of the segmentation (width, height, and depth) into vectors representing the 3D-dimensional cartesian coordinates [X,Y,Z], and their respective intensities into an additional vector (*PCBuilder*).

## Projections

To quantify tissue spreading over the full surface of the fly embryo, we used cylindrical projections as described (*Krzic et al., 2012*; *Rauzi et al., 2015*). Briefly, the anterior-to-posterior axis of the egg was calculated (*CovMat3D*) and aligned along the Z axis of the coordinate system (*VecAlign*). The cartesian coordinates were then transformed into cylindrical coordinates $[X,Y,Z]->[\theta,r,Z]$ (*Cart2Cyl*). For each position along Z and from 0 to $2\pi$ along $\theta$, the mean intensities of all points between $r_{max}$ and $r_{min}$ were projected as pixels along width and height [W,H] of a two-dimensional image I (*CylProjector*). Translations and rotations (*PCRotator*) in the cartesian point cloud were used to position biological landmarks (e.g. dorsal/ventral midline) in I. To allow for a mapping of information obtained in I (tissue areas and cell tracks) back to the point clouds and stacks, the index information of all projected points was also stored in a vector array. Our cylindrical projection provided an approximate area conservation in the central domain of the embryo that was sufficient for visualization purposes. For quantitative analyses of serosa expansion, distortions were corrected at poles and furrows by using the law of cosines to weight the area of each pixel in I according to its contribution to the corresponding surface voxel in the embryo.

## Membrane segmentation

To quantify main aspects of cell shapes in fixed tissue, Phallacidin stained cells were segmented semi-automatically using Ilastik (Linux version 1.2.0) Pixel Classification framework (*Sommer et al., 2011*). In the case of mis-segmentation by the automatic algorithm due to lower resolution, cell outlines were corrected manually. Predictions were exported as a binary image stack. The spatial position of each cell within the imaged volume was defined by the centroid of the segmented cell. Individual cell volumes were extracted as a single connected component, the resulting objects were loaded as point clouds into Matlab and remaining holes were closed using *fillholes3d* with a maximal gap of 20 pixels (iso2mesh toolbox). To account for possible artifacts in image processing, objects smaller than 200 pixels and larger than 10000 pixels were excluded from further analyses. To reveal

changes in cell and tissue dynamics in time-lapse recordings, individual cells and the expanding serosa were outlined manually.

## Feature extraction and quantification

Cell height was measured as object length orthogonal to the embryo surface. Cell surface area and cell circularity were measured by a 2D footprint that was obtained through a projection of the segmented cell body along the normal axis of the embryo. Cell tracks were obtained manually by following cells in selected layers of cylindrical projections. Germband extension was measured in midsagittal sections of time-lapse recordings: the most anterior point of the dorsally extending germband was used as reference, and germband extension was measured in percent egg length relative to the anteroposterior length of the embryo.

## Cross correlation

A quantification of substrate and membrane movements was obtained using Matlab's Computer Vision System Toolbox implementation of the Lucas-Kanade optical flow method on the respective layer projections. The orthogonal component corresponding to the AP axis was analysed on manually generated single cell segmentations, evaluating the area around the cell outline for the fluorescent signal of Lifeact-mCherry, and the entire cell area for the fluorescent signal of *Mab-bsg-eGFP*. The mean of the magnitudes was calculated for each frame and the cross correlation of both resulting vectors was evaluated for all 200-s time windows as described using Matlab's Signal Processing Toolbox cross-correlation function (*Goodwin et al., 2016*). Negative controls were realized by calculating correlation between randomized cell and substrate area.

## Statistics

Statistical comparisons of correlation coefficients distribution were performed via Matlab's implemented $\chi^2$-square test for normal distribution, Student's t-test (two sided, unpaired), and hample's test. All samples fit normal distribution; P-values of Student's t-test are indicated in the figure legend. No statistical method was used to predetermine sample size. The sample size (N) for the groups was decided based on current literature for developmental biology (e.g. *Goodwin et al., 2016*). Each embryo or cell is considered a biological replicate. The size of N (embryos and cells) is indicated on each figure.

## General image processing

3D reconstruction images of individual cells from z-stack segmentation data were done in Matlab (R2016b), images and stacks were processed using Fiji (2.0.0-rc-34/1.50a) and Matlab, and panels were assembled into figures in Adobe Photoshop and Adobe Illustrator. Custom Matlab functions for SPIM data processing are indicated with capital first letter and are available via github (*Gonzalez Avalos, 2018*; copy archived at https://github.com/elifesciences-publications/SPIMaging).

## Acknowledgements

We thank A Guse and N Bloch for help with establishing a protocol for recombinant Lifeact-GFP; P Lenart for suggesting the use of labeled H1 and mRNA polyA-tailing; K Goodwin and G Tanentzapf for helpful suggestions on the cross-correlation analyses; M Benton, L Centanin, N Gorfinkiel, A Guse, K Panfilio, U Schmidt-Ott, J Wittbrodt, and members of the Lemke lab for discussions and/or comments on the manuscript. We acknowledge K Panfilio and two anonymous reviewers for their constructive feedback; I Lohmann, J Lohmann and J Wittbrodt for sharing laboratory equipment, and A Maizel for support. We are grateful to J Wittbrodt for his continuous and generous support. Funded by DFG grant LE 2787/1–1, HFSP grant RGY0082/2015, and a pre-doctoral HBIGS fellowship to LS.

## Additional information

### Funding

| Funder | Grant reference number | Author |
| --- | --- | --- |
| Deutsche Forschungsge-meinschaft | LE 2787/1-1 | Francesca Caroti<br>Paula González Avalos<br>Maike Wosch<br>Steffen Lemke |
| Human Frontier Science Program | RGY0082/2015 | Everardo González Avalos<br>Viola Noeske<br>Paula González Avalos |
| European Molecular Biology Laboratory | EMBL International PhD Programme | Dimitri Kromm |
| Center for Modelling and Simulation in the Biosciences | | Lars Hufnagel |
| Universität Heidelberg | Heidelberg Biosciences International Graduate School Predoctoral Fellowship | Lucas Schütz |

The funders had no role in study design, data collection and interpretation, or the decision to submit the work for publication.

### Author contributions

Francesca Caroti, Conceptualization, Formal analysis, Investigation, Methodology, Establishment of SPIM for injected Megaselia embryos, Writing—original draft, Writing—review and editing; Everardo González Avalos, Conceptualization, Resources, Software, Formal analysis, Investigation, Visualization, Methodology, Writing—review and editing; Viola Noeske, Conceptualization, Formal analysis, Validation, Investigation, Writing—review and editing; Paula González Avalos, Formal analysis, Investigation, Validation, Visualization, Writing—review and editing; Dimitri Kromm, Formal analysis, Investigation, Methodology, Writing—review and editing; Maike Wosch, Resources, Methodology, Writing—review and editing; Lucas Schütz, Resources, Data curation, Writing—review and editing; Lars Hufnagel, Supervision, Resources, Methodology, Writing—review and editing; Steffen Lemke, Conceptualization, Formal analysis, Supervision, Funding acquisition, Validation, Investigation, Visualization, Writing—original draft, Writing—review and editing

### Author ORCIDs

Steffen Lemke (iD) http://orcid.org/0000-0001-5807-2865

### Decision letter and Author response

Decision letter https://doi.org/10.7554/eLife.34616.021
Author response https://doi.org/10.7554/eLife.34616.022

## Additional files

### Supplementary files

• Transparent reporting form
DOI: https://doi.org/10.7554/eLife.34616.019

### Data availability

All data generated or analysed during this study are included in the manuscript and supporting files. Custom Matlab functions are available via GitHub (https://github.com/lemkelab/SPIMaging; copy archived at https://github.com/elifesciences-publications/SPIMaging).

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
