## [Decision Letter]

Thank you for submitting your article "In toto live imaging in scuttle fly *Megaselia abdita* reveals transitions towards a novel extraembryonic architecture" for consideration by *eLife*. Your article has been reviewed by three peer reviewers, and the evaluation has been overseen by a Reviewing Editor and K VijayRaghavan as the Senior Editor. The following individual involved in review of your submission has agreed to reveal her identity: Kristen A Panfilio (Reviewer #2).

The reviewers have discussed the reviews with one another and the Reviewing Editor has drafted this decision to help you prepare a revised submission.

All three reviewers agree that the manuscript addresses the very interesting question about the nature of morphogenetic transitions accompanying evolution, using the comparative analysis of the development of insect extraembryonic membranes as a paradigm and found the conclusions presented in the manuscript both intriguing and thought provoking. Specifically, the suggestion that the morphogenetic transitions arose, not from a change in the transcriptional differentiation programs that affect fate, but rather from a genetic change that modulates extra embryonic tissue interactions is very interesting. However all reviewers had substantial concerns about the strength of the evidence for some of the conclusions drawn and found the arguments hard to follow. I summarise the major concerns below.

i) The premise and the evidence to suggest that the evolutionary change accompanying EE morphogenesis is gradual rather than sudden is poor.

ii) The suggestion that the modification of inter-tissue (inter EE membrane) interactions contributed to the differences in morphogenesis is the most interesting conclusion of the manuscript. However, the conclusion is neither substantiated by the visualization of these interactions/ topologies, nor validated functionally. Transverse sections from fluorescently stained embryos or EM sections and the analysis of ECM components in wildtype and mutant development will help strengthen this conclusion.

iii) The analysis of cytoskeletal, cell and tissue dynamics, including actin based protrusions, cell shape changes, cell spreading and what the authors describe as 'flipping over" is not of sufficiently high resolution and based on a small sample size.

iv) The comparative analysis between the three modes of development (ancestral, intermediate and derived) can be very valuable but it is not clear from the data presented whether the developmental stages being compared are appropriate.

v) The manuscript is, as it is presented, very descriptive and hard to understand. It could be improved in clarity through the use of schematic representations of the comparative anatomy/ topology of the EE membranes and of timelines indicating the major events accompanying the morphogenesis of the EE membranes.

vi) Both the images and the videos lack sufficient detail in the legends with respect to what is being shown and will also benefit from better labeling.

vii) As outlined by reviewer 2 in the detailed comments below, the controls used in the genetic manipulation experiments (RNAi, CRISPR/Cas) should be clearly documented, and the key defining features of wild type serosal and amniotic cellular properties starting at blastoderm stages (Figure 2) must be derived from more than a single biological specimen.

The detailed reviews of all three reviewers are pasted below.

*Reviewer 1:*

In this manuscript, Lemke and colleagues investigate the basis of evolutionary transitions in morphogenetic movements using the early development and dynamics (blastoderm to germband retraction) of the extraembryonic (EE) tissues in the scuttle fly, *Megaselia abdita* as their model, on account of its intermediate mode of development of these structures compared to the ancestral (typified by the flour beetle, *Tribolium*) or the more derived mode (seen in the fruitfly). Using live imaging (SPIM) to capture the dynamics of the EE tissues and the quantitative analysis of tissue dynamics and cell behaviours, they show that the *Drosophila* amnioserosa more closely resembles the *Megaselia* serosa and suggest that the derived form of development in the former resulted from two morphogenetic changes, the loss of fusion of the ventral amnion (seen in the ancestral mode) resulting in an open amnion in *Megaselia* and the loss of serosal spreading in the derived mode in *Drosophila* (but observed in *Megaselia*).

To corroborate their conclusions, the authors use live imaging to track tissue and cell dynamics and show that although the fixed preparation morphologies of the amnion and serosa cells in early *Megaselia* development are indistinguishable, their dynamic properties are. Specifically, they show that the serosa cells spread, and pulsate while the amnion cells do not exhibit changes in cell shape or spread. Further they show that the serosa contributes to germ band retraction, the amnion does not and remains open ventrally even after germ band retraction. They compare the cell and tissue dynamics associated with the two tissues in *Megaselia* with that of the composite amnioserosa in *Drosophila* and argue that the latter more closely resembles the serosa albeit one that has lost its ability to spread. Finally they show that knocking down an extracellular matrix (ECM) regulator, *mmp1*, affects serosa dynamics and adhesion to the yolk in *Megaselia* resulting in a morphology similar to the amnioserosa but does not affect either germ band extension or retraction. They suggest that the yolk sac is a neglected contributor to EE morphogenesis.

The manuscript addresses the interesting issue of the nature of evolutionary transformations that accompany evolution and suggests that the changes in extraembryonic membrane dependent morphogenetic events arose, not from a change in the transcriptional differentiation programs that affect fate but rather from a genetic change mediated by a matrix metalloprotease, that modulates extra embryonic tissue interactions with the yolk sac. Overall, I found the conclusions presented in the manuscript both intriguing and interesting. I did however find some of the evidence presented and the conclusions drawn, hard to follow. I list my concerns below.

i) The dynamic attachment and/or detachment between the extraembryonic membranes in the wildtype is not easily discerned in the videos or images presented. Could the authors label one but not the other layer and use nuclear markers to enable the clear visualization of both layers? Alternatively, can the authors show images and snapshots from videos in which the outlines of cells of the two layers drawn out (after one layer has been digitally peeled off) and projected to show how these associations (and shapes) are dynamically changing? Also, orthogonal or real transverse sections of the anterior, middle and posterior regions from Lifeact/phalloidin and DAPI- labeled embryos at different stages might help illuminate the topologies of interacting interfaces better.

ii) The stages at which cell and tissue morphogenesis between *Drosophila* and *Megaselia* are being compared is not very clear in the data represented in Figure 6. Could the authors include a Materials and methods section that describes staging? This is particularly important as most of the dynamic analysis of the *Drosophila* amnioserosa has been done at and after the onset of germband retraction.

iii) The dynamic interactions between the extraembryonic membranes and the yolk sac, both in wildtype and in the *mmp* mutant embryos is not clear. Demonstrating this will be particularly interesting in light of the fact that the yolk cell has been shown to be necessary for both germband retraction and dorsal closure in *Drosophila* through its interactions and close apposition with the amnioserosa. Here, integrin laminin and basigin have been shown to mediate their interactions. Can the authors examine the deposition of laminin at different stages and their alteration in *mmp* mutants? Alternatively, and especially as *mmp* mutants do not seem to affect the morphogenetic process, could the authors examine the distribution and effects of the loss of one of the three components mentioned above?

iv) Schematic cartoons and timelines describing the major events accompanying the early morphogenesis of the extra embryonic membranes will aid a better understanding of the manuscript which is otherwise very descriptive.

*Reviewer 2:*

This manuscript examines early morphogenesis of the extraembryonic (EE) epithelia in the scuttle fly *Megaselia abdita*. This is presented as an "intermediate mode of EE tissue formation" compared to a more ancestral situation, as seen in the beetle *Tribolium castaneum*, and the derived situation seen for the single amnioserosal tissue in *Drosophila melanogaster*. Although the description of *Megaselia* EE morphogenesis is very detailed and involves technically sophisticated and diverse analyses, I have major concerns with the datasets, their analyses, and the conceptual framework of the current manuscript. In order to allow the authors sufficient time to address these concerns on a potentially very interesting topic, a new submission after extensive revision would be appropriate. Alternatively, analytical and presentation revisions would make the current manuscript potentially suitable for a journal that focuses specifically on developmental biology.

Briefly, my major concerns are stated in the topic sentence of each of the following paragraphs and are pasted here for quick reference:

1) I found the polemical framework of saltatory versus gradual evolutionary change to be rather muddled and of questionable relevance.

2a) The significance of the findings is not very strong (for conceptual advance or for strength of the results underpinning certain conclusions).

2b) The authors posit a number of intriguing aspects regarding tissue interactions, but they lack the resolution and the functional testing that would be required to support such suppositions.

3) The sample sizes are alarmingly small.

4) The use of an F-Actin fluorescent label to characterize cell shapes provides only limited resolution, and I cannot clearly see a number of features described by the authors.

5) These experimental design limitations seem to carry over into the analyses in several places.

Major concerns in detail:

I found the polemical framework of saltatory versus gradual evolutionary change to be rather muddled and of questionable relevance. Anecdotal transition state fossil evidence such as for *Archaeopteryx* seems to be a weak basis to assume that the evolution of epithelial morphogenesis must occur gradually. Developmental characters such as the presence or absence of abdominal appendages in arthropods provide a clear example of specific genetic changes having fairly abrupt, nearly binary morphological outcomes. Even when I try to disentangle the authors' ambiguous usages of the term gradual, which seems to either refer to evolutionary change or to developmental temporal progression (both meanings seem relevant in the middle of the penultimate Discussion section paragraph), I do not see a reason to posit its relevance here. For example, unless inter-embryo phenotypic variation is seen for the RNAi and CRISPR *mmp1* experiments, it appears that the serosa either spreads or does not, but no intermediate levels of spreading are shown. Also, I would caution the authors that developmental genetics investigations in extant species does not really provide a forum for "experimental evolution" and that development in *Megaselia* does not constitute an ancestral or transition state.

The significance of the findings is not very strong. In the Introduction the reader is informed that the "absence of a ventrally closed amnion" is "characteristic for this intermediate mode" seen in *Megaselia*. Why, then, is it "surprising" in the third Results section to find a ventrally open amnion? I was, however, surprised that the authors do not directly address how their data support a major topographical revision of the *Megaselia* amnion compared to the many cited publications that presented this tissue as being dorsally closed, over the yolk. Rather, a very thin tissue region with a morphologically distinct cell type that forms a *lateral* ring around the embryo seems to be very novel – not something that can readily be homologized with the ventrally-closing amnion of *Tribolium*. In general, I would ask the authors to consider the bigger picture context of mature tissue physiological function in assessing potentially homologous characters of the EE tissues, such as serosal cuticle secretion (or other protective and physiological roles performed by a completely enclosing serosal sack) or formation of a sealed, fluid-filled amniotic cavity. In several places in the manuscript, I also found interspecies comparisons specifically for morphogenetic features to be rather casual and vague.

Beyond these issues of interpretation, the authors posit a number of intriguing aspects regarding tissue interactions, but they lack the resolution and the functional testing that would be required to support such suppositions. This includes descriptive, correlational observations for the claims that (a) serosal spreading is autonomous and active (so what's distorting the serosa in the *mmp1* knockdowns?), (b) there is a yolk membrane that experiences differential adhesion to the serosa over time (not all insects have a yolk membrane and it can vary across species in structure), (c) that oscillations within the serosa are mechanically relevant for serosal detachment. Drug treatments or laser cutting could address some of these mechanical predictions, while high resolution visualization in fixed specimens (TEM and/or immunohistochemistry) would clarify ECM structure and the nature of potential cellular junctions at tissue interfaces.

The sample sizes are alarmingly small. I fully appreciate the big data and computational demands involved in light sheet microscopy work, but some key observations are based on analyses of only a handful of cells from a single biological specimen. Nowhere in the figures, videos, or analyses did I see use of the histone-labeled, lower magnification recordings, which cannot visualize cell outlines and already curtails the effective dataset by 25%. Furthermore, not all recordings span the same developmental stages: the key defining features of wild type serosal and amniotic cellular properties starting at blastoderm stages (Figure 2) therefore derive from a single biological specimen, in which only 23 cells were examined quantitatively (it is unclear whether this value is even per tissue type or in total across three distinct tissues). Even for developmental stages documented in multiple recordings, there is only a sample size of 2 – or in one case 3 – individuals, and most quantitative plots are for fewer than ten cells (unclear if multiple specimens were used in such analyses, e.g., Figures3B, 6D, 6I, 7H). For example, is the sequence of serosal detachment (anterior, posterior, then lateral) consistent across specimens, and what is the error or variability in calculations of tissue area increase over time?

The use of an F-Actin fluorescent label to characterize cell shapes provides only limited resolution, and I cannot clearly see a number of features described by the authors. In actively remodeling tissues, F-actin is not the most reliable indicator of cell shape. Since labeling was achieved via injection of an existing construct, why was a membrane label, such as Gap43-YFP not used additionally or instead? Even in frame-by-frame inspection of the manuscript's videos, I find it very difficult to fully discern individual cell outlines at all time points or to see the nature of the tissue organization dorsal to the amnion (in peeled off serosal renderings). The cut-away volumetric renderings are helpful, but it does not appear that F-Actin clearly or fully label the basolateral domains of cells, which is critical for ascertaining cell shape and volume. The use of a ubiquitous label also precludes rigorous tissue discrimination and cannot distinguish apical-basal polarity, which seems crucial, for example, in confirming the authors' claims that amnion cells "flip over". Moreover, F-Actin does not label the yolk sac.

These experimental design limitations seem to carry over into the analyses in several places. There are several imprecise renderings of image data that impair clarity and do not create a strong impression of robust analyses for a study that is largely based on manual tracking. For example, there is a discontinuity in labeling (and ability to identify?) amnion cells in Figure 1F-I (why is there a gap posterior to cell #2?). Furthermore, tinted regions to indicate cells and tissues seem somewhat arbitrary or imprecise with respect to the F-actin signal (Figures 3C and 4D, and Video 6). In Video 6, the tinting to indicate the entire serosal tissue is markedly inconsistent between consecutive time points regarding which cells are included, and the tinted region often does not respect cell boundaries. This seems to be in contrast to consistent tissue edge shapes over time as shown in Figure 5B. Furthermore, the patchy, limited scope of wild type *Mab-mmp1* expression seems to contradict a role in a membrane layer at the yolk's surface. Also, can the authors reconcile the lengths of protrusions observed by live imaging (which are very hard to see) with that documented by expression of *Mab-egr* transcript (compare Figure 3D-E with 3G')?

Specific methodological concerns:

How many RNAi and CRISPR/Cas experiments were performed (number of instances of injection)? Were multiple, non-overlapping dsRNA fragments used for *Mab-mmp1* to ensure specificity of knockdown (avoid off-target gene knockdown effects)? Can the authors comment on the efficacy of CRISPR/Cas with respect to biallelic editing or the degree of mosaicism? It may be that failure of serosal spreading is a generic phenotypic outcome when embryos are unhappy: what negative control (i.e., targeting a completely different gene) was performed?

Given that oscillations in apical area are the key distinguishing feature between the *Megaselia* amnion and either the *Megaselia* serosa or *Drosophila* amnioserosa, please strengthen the quantitative presentation of this. What is the periodicity, duration of the oscillatory phase, developmental stage of onset and cessation, and level of inter-embryo variability? Why were other quantified features such as apical area and apical circularity not also applied in characterizing the amnioserosa?

As injection and fluorescent live imaging can interfere with embryogenesis, what controls did the authors take to ensure that these treatments did not impair healthy development?

Insect development is temperature dependent, yet many events are presented in terms of absolute time in minutes. Please take care to note the temperatures for incubated eggs (such as prior to in situ hybridization) and during live imaging recordings, and to relate these clearly to developmental stages.

As detailed as the Materials and methods section is, I do not find the light sheet and point cloud sections sufficient to be reproducible. Actions such as "smoothening" and "eroding" are only informative in light of the operating system/ software/ custom scripting/ other, which needs to be cited and or made available (e.g., via GitHub).

Presentation concerns:

The entire paper would strongly benefit from a substantially strengthened explanatory framework for *Megaselia* development, with: schematic representations of sampled regions over the egg circumference and of specific cell shapes and tissue boundary organizations; a clear statement of developmental staging and landmarks; and explicit definitions of quantified features. For example, when exactly does serosal spreading initiate and complete relative to germband extension and retraction? How exactly was apical circularity calculated and what is the "ideal circular apex"?

Across both figures and videos I found serosal separation from the blastoderm impossible to see clearly. How was this determined kinematically, and why is there ambiguity about this in the oldest stage analyzed for Figure 1? Similarly, given the 4D reconstructions from SPIM data, why is no ventral view to illustrate serosal closure provided?

Color legends and axis labels are missing in several cases and/or are not defined in the figure legend. For example, what is the primary y-axis in Figures 1E-G, 3B, 5D-E, and what does "GBE [%]" mean as a y-axis label in 6E, 6J, 7I (given that 100% GBE is never achieved)? What is the secondary y-axis label in 2J and 3B? Please improve the text resolution for in-figure labels of samples sizes, and please explicitly indicate how many biological specimens and tissue types are included in any given n. For the very limited sample sizes, please provide (schematic) insets to show where along the embryo surface those cells are located. Please define colors such as for the heat maps in 5D, 6C, 6H, 7F (is darker older or younger?), and the various use of pinky-purple in plots compared to blue-turquoise for cell traces (6D, 6I). What are the heat maps on certain plots (5C, 6B, 6G, 7G)?

There appears to be a displaced box for the ROI in Figure 1A: the actual cells shown at high magnification (1B-C) are just posterior to the position of the box in 1A.

What are the error bars in Figures 2J and 3B?

Presumably the first time point in Figure 3D-E is magenta, not cyan.

Presumably the images in Figure 3D-E are shown in dorsal-flattened aspect with anterior up and centered on the dorsal midline (constituting a 90-degree rotation relative to the images in 3F-G), but I should not have to guess these things!

There is no figure legend for Figure 6 panel E.

Please take care to distinguish an entire egg/ biological specimen from embryonic tissue proper. Using "embryo" to refer to both is very confusing, particularly for blastoderm stage events.

Please be consistent in nomenclature for species (*M. abdita* or *Megaselia*) and always italicize.

I have no stylistic objections to the use of more informal language so long as meaning is clear. But why not use "columnar" and "squamous" as appropriate to define cell shape?

Several brief sections were not clearly introduced and their relevance and significance were not apparent, such as the exploration of the anterior amnion and the very detailed analysis of zebrafish EVL expansion.

*Reviewer 3:*

The authors present an important and interesting study of extraembryonic tissue morphogenesis in non-model insect species *Megeselia abdita*. Using quantitative description of cell shapes and behaviours in fixed and SPIM imaged *Megesalia* embryos they paint a complete picture of extraembryonic tissues spreading in this species. Specifically, they show that *Megaselia abdita* amnion remains an open tissue and that the serosa cells show area pulsations similar to *Drosophila* amnioserosa. In addition, genetic perturbation experiments in *Megaselia* provide evidence for coupling of the serosa to the underlying yolk. Throughout the paper, the experiments in *Megaselia* are compared and contrasted to the wealth of data on embryonic extraembryonic relationship in two more traditional insect models – *Drosophila* and *Tribolium*. The comparative approach gives the paper its structure and underlying theme, asking an important question of how gradual changes in cell dynamics lead to the formation of highly divergent tissue level behaviour, however appears largely speculative.

Introduction is well written and clear. However, the model system used is relatively unknown and a schematic of the anatomy and current knowledge of the dynamic tissue remodelling events would help the reader orient in the subject. Such schematic should then be used consistently in most of the following figures that are hard to follow without conceptual visual aid.

In their first experiment, the authors quantified cell shape in fixed stained *Megaselia* embryos at several stages of development. The data are clear when it comes to apical cross section of the cells. From the quality of the images, even in the zoomed-in versions, it is less clear how accurate could the automated segmentation of cell shapes be in the axial direction (along the apical basal axis of the cells) where the resolution must be necessarily significantly lower. Could the authors show in supplements such cross sections and comment on the reliability of their Ilastik segmentation pipeline on this data? The PCA analysis is summarized in many plots that are largely redundant.

Tracking experiments in SPIM recordings show convincingly that serosa cells are able to spread over adjacent cell layers and that these cells can be mapped onto the region of the blastoderm from which serosa originates (based on zen expression – Figure 2E could use some context). The cells next to the serosa cells are identified as amnion based on their size and cell division status and the fact that they are overgrown by serosa and eventually separate from it. In the absence of clear genetic evidence for the mapping of these cells to amnion, it would be prudent to refer to them as "putative amnion" rather than "amnion" as the authors chose to do in the rest of the paper. It is unclear how the last paragraph of this section tests the robustness of tracking. This confusion stems perhaps from the lack of clear conceptual explanation of what the serosa amnion separation means and what behaviour the amnion cells are expected to show.

In the next set of analyses of the SPIM movie the authors follow the ultimate fate of the putative amnion cells. The section is rather confusing. On the one hand, authors are surprised that the lateral amnion remains ventral and yet they seem to expect that the amnion will close ventrally (first paragraph). The *Mab-eiger* staining reveals, presumably viewed from the dorsal side, that amnion does not close dorsally and that the cells have protrusions. I appreciate the difficulty of visualizing the live in toto imaging, however the statements about cells "flipping over" are completely unclear from the data. Also, the evidence for the filopodia like protrusions is not convincing from the live imaging data. Given these difficulties in understanding the text and the data, the strong evolutionary conclusion about "gradual transition in the evolution from the ancestral ventral amnion" cannot be easily accepted. I propose a thorough rewrite of this section and to enhance the data with conceptual drawings for further clarity. To substantiate the "flipping", is it possible to confirm the apical-basal side of these cells with an apical polarity marker?

The authors next describe a supracellular actin cable between serosa and amnion. While I believe that such a cable exists, the pictures in Figure 4 do not show it very clearly. The lines mostly obscure the presumed accumulation of actin. Please show the panels with and without the lines. At the end of the second paragraph, authors state that "the serosa continues to spread laterally and eventually broke free, leaving behind a single row of amnion cells […] the actin cable was observed in the expanding serosa but not the remaining amnion." This seems important and would be worth substantiating with some panels in the figure.

Next the authors discovered pulsations in the extraembryonic tissue. They claim that pulsations were observed only in cells of the serosa that are not participating in the cable. To me the data in Figure 5D, E show that both border and non-border cells pulsate, while border cells perhaps less. A statistical treatment of the data is required. Moreover, authors speculate that the pulsation decouples the serosa from the yolk sac. Several questions arise. Apparently, the increased cell oscillations coincide with the pulsation of the yolk. Are these pulsations in the yolk cells? Please show the yolk pulsations data. What indicates that the pulsations decouple the serosa from the yolk?

The authors next turn to genetic perturbation experiments. These are non-trivial experiments in *Megaselia*, in particular when coupled with complex SPIM imaging. In Zen RNAi serosa is transformed into amnion and accordingly there is no spreading. Authors report lack of pulsatile behaviour that they interpret as amnion-like. In Figure 6D, I lack quantitative comparison of the extent of pulsatile behaviour to either serosa or amnion wildtype cells. In *Drosophila*, amnioserosa cells do not spread but pulsate extensively. Authors confirm this with their own in toto recordings. I am not sure what can one conclude from comparing the germband retraction dynamics of wildtype *Drosophila* and Zen-RNAi *Megaselia*. Therefore, the evolutionary statement at the end of the paragraph remains highly speculative.

I find the final experiment of the paper most impressive. Knocking down extracellular adhesion molecule *mm1* in *Megaselia* prevents serosa spreading, the embryos strikingly resemble *Drosophila* and yet development is unaffected. Analysis of SPIM recordings shows pulsations however it is claimed that it is dampened. Again, this claim needs to be better substantiated. The connection between serosa cells and yolk, proposed two sections ago, gains some traction here and in fact should be presented together.

Overall, I find the paper thorough, thought provoking and valuable to the scientific community. The overall pitch towards the fundamental question of the gradual evolution of developmental mechanism seems a little bit exaggerated. It belongs to the Discussion and should not drive the narrative of the paper. The data simply do not (and cannot) support broad statements such as 'amnioserosa in *Drosophila* evolved by loss of serosa spreading without disrupting the developmental programs of serosa and amnion'. This does not mean that the authors cannot speculate in that direction. A clear conceptual model figure at the end would highlight the exciting evolutionary implications of this work.

[Editors' note: further revisions were requested prior to acceptance, as described below.]

Thank you for resubmitting your work entitled "Decoupling from yolk sac is required for extraembryonic tissue spreading in the scuttle fly *Megaselia abdita*" for further consideration at *eLife*. Your revised article has been favorably evaluated by K VijayRaghavan (Senior Editor), a Reviewing Editor, and three reviewers.

The manuscript has been improved but there are some remaining issues that need to be addressed before acceptance, as outlined below:

As you can see, the authors need to address straightforward matters. This can be speedily done. We append the reviews in full (there is much overlap between 1 and 2. Please address these so we can proceed for acceptance,

*Reviewer #1:*

This is a revised version of manuscript by Lemke et al. In the earlier version, they used the early development and dynamics (blastoderm to germband retraction) of the extraembryonic tissues in the scuttle fly, *Megaselia abdita* (as a model for the intermediate mode of development of these structures) to compare it with the ancestral mode (typified by the flour beetle, *Tribolium*) and derived mode (seen in the fruitfly) and argued that the *Drosophila* amnioserosa more closely resembled the *Megaselia* serosa. They also suggested, based on these findings that the derived form of development resulted from changes that involved the tinkering of morphogenesis, specifically, tissue interactions rather than changes in gene regulatory networks that govern them, and invoked the loss of fusion of the ventral amnion and the loss of serosal spreading to explain the basis of the derived mode observed in *Drosophila*. The revised version makes one major conclusion, namely that the "decoupling" between the yolk sac and serosa, regulated by *Mmp1*, contributes the fast phase of serosal closure in *Megaselia*. They use this finding to explain (in part) the faster rate of serosal closure in *Megaselia* compared to dorsal closure (in which zipping and fusion slow it down further).

To corroborate their conclusions, the authors use nuclear and membrane markers and live imaging using SPIM to show that the dynamic properties of the amnion and serosa are distinguishable enough to allow area based tracking. Using this, they first confirm the fate map of the *Megaselia* blastoderm with respect to the cells that contribute to the amnion and serosa and then resolve their dynamic topologies using image-processing routines that rebuild 3D projections from flattened topologies. Specifically, they focus on the cellular topology of the amnion cells during and after their disjunction from the serosa (which they do not describe or discuss in detail) and then on the dynamics of the serosa. Their findings corroborate earlier reports on serosal dynamics but additionally uncover a phase when serosa cells appear to pause. This phase, that separates the fast and slow phases of serosa closure, they argue, results from the mechanical coupling to the yolk. To substantiate this, they show that the serosa (visualised using Lifeact GFP) and the yolk cell (visualised using a Basigin GFP injected into the yolk cell) exhibit coordinated movement or oscillations (optical flow and cross correlation analysis) in the slow phase but become less coordinated/ correlated in the fast phase. They use CRISPR/RNAi to show that this "decoupling" of the yolk and serosa result from the MMP mediated regulation of their interactions. In *Mmp* mutants, they show that the fast phase does not occur and the correlated dynamics persist. They use these findings to suggest that loss of serosal spreading in *Drosophila* may result from modifications in interactions between the yolk and (amnio)serosa.

Overall, I think the manuscript makes a convincing case for the role of *Mmp1* in regulating serosal closure dynamics. While there is clarity with respect to the dynamics of the amnion, I am still somewhat confused about the dynamics of the serosa, specifically, its dynamic relationship to the yolk cell and the amnion, (which forms the center-bit of this manuscript) and what *Mmp1* actually does to the interactions between the extraembryonic tissues. I am also not sure whether serosal closure rates can be compared to dorsal closure rates. I have the following major questions:

i) How does "disjunction" correlate (temporally) with serosal spreading? Could this explain the differences in serosal dynamics between the serosa and the amnioserosa (composite)? What does "disjunction" involve: loss of adhesion between the amnion and serosa? Does *Mmp* also affect this?

ii) Does "decoupling" also imply physical detachment between the yolk and the serosa? While the authors argue for decoupling based on correlating movements, it will be good if they can show both in the transverse sections and schematics of the kind shown in Figures 2G and 2A respectively, the positions of the yolk and serosa when they are decoupled. Is the dorsal most serosa still sitting on the yolk while the lateral serosa is not? Confocal real time movies focusing on the midsaggital plane will also be useful to examine the former.

iii) Are the authors suggesting that the fast phase of serosal closure uses the epidermal cells as a substrate for spreading? What happens to the epidermal cells during the process? Are they moving dorsal ward along with the amnion? How does the fast phase of serosal closure correlate (temporally) to the anteriorward movement of the epidermis or the contact of the serosa with the epidermis?

v) A timeline indicating the major events involving the extraembryonic membrane including germband extension and retraction and the morphogenetic transitions in the three tissues including disjunction, decoupling, fast, slow and paused phases of serosal dynamics marked on this timeline would be useful. This is particularly important as the manuscript examines the dynamics of the tissues across different morphogenetic movements (extension and retraction of the germband).

*Reviewer #2:*

The manuscript has been extensively revised and is much improved for clarity of text, figures, and videos, and for the thematic focus on extraembryonic tissue morphogenesis. I have the following comments to improve documentation and clarity.

1) A crucial feature of the study is the disjunction of the serosa from the adjacent blastodermal epithelium. This is an essential prerequisite for the serosa to spread over and enclose the embryo, and several analyses in the manuscript distinguish morphogenetic behaviors before and after disjunction. How is disjunction ascertained? Nowhere did I find a clear statement of how this was determined. There may also be some confusion on this point, as it is difficult to understand how the serosa could exert displacement drag on the amnion after they have separated (subsection “The amnion in *M. abdita* develops as a lateral stripe of cells”, last paragraph). Note that qualitative descriptions, such as of the serosa 'breaking free' (subsection “*M. abdita* serosa spreading proceeds in distinct phases”), carry mechanical implications that are not necessarily justified.

2) Amnion morphogenesis: The supplementary videos and figure associated with main text Figure 2 are excellent, and clearly show the unusual behavior of the lateral amniotic cell row folding over along the dorsal-ventral axis. Does this folding over occur in an anterior-to-posterior progression (the small region shown in the videos suggests this might be the case)? However, at later stages while I can see the dorsally protrusive cell shape changes, I am less convinced that the cells exhibit crawling behavior: (a) the ventral actin cable that delineates the amnion from the epidermis remains quite distinct and straight, suggesting the retention of tissue-level organization, and (b) my impression from the video is that the amnion is rather displaced passively due to other developmental events, while crawling implies active behavior. Please clarify the description of the amnion for these features.

3) The new investigation of serosal-yolk sac interactions is now the centerpiece of the manuscript, emphasized in the revised manuscript title. Given the importance of these experiments, please clarify the following:

A) Although perhaps less important than fluorescent reporter localization, the in situ characterization of endogenous *Mab-basigin* would be an important aspect of characterizing a new gene and labeling tool in *Megaselia*. Unfortunately, the in situ micrograph is neither convincing nor informative: potential yolk sac signal seems no stronger than potential background signal in the head. In the main text there is a comment about delaying injection of the Basigin label so as to restrict localization. Please reconcile statements on the level of specificity of basigin and of the transgenic label (subsection “De-coupling of serosa and yolk sac is preceding serosa tissue spreading”, second paragraph). For endogenous transcript, sectioned rather than whole mount material would clarify the extent to which expression is restricted to the yolk periphery, even with coarse sectioning by hand.

B) For the material analyzed in Figure 4, where is this in the embryo? Morphologically, it looks as though the epidermis is included in the lower part of the viewing field (Figure 4A). An inset schematic would help for orientation. In the accompanying video, I find it hard to understand the spatial and temporal context of the video on the right. Why are the viewing fields in the video cropped relative to the main text image, and why do they differ between the left and right videos within the file?

C) More generally, spatially and across embryos, how representative is the analysis in Figure 4? It currently appears to report a detailed analysis of only a single, small tissue region in a single specimen. While I note sample sizes within Figure 5, this information could be made more explicit in the main text, along with the spatial information.

D) The legend to Figure 4 indicates that a t-test was used in assaying serosa-yolk correlation before and after disjunction, but are the data normally distributed such that the assumptions of the t-test are met? (There seems to be a skew towards higher correlation values for the data shown in Figure 4F.)

E) Subsection “De-coupling of serosa and yolk sac is preceding serosa tissue spreading”, last paragraph: why "presumably" in describing potential oscillations? Can these oscillations be linked to biological processes such as serosal spreading or events in the embryo such as segmentation or germband expansion? Given that they serve as a landmark feature in subsequently characterizing the mmp1 loss-of-function data, a brief comment on this developmental feature would be appreciated.

4) Comparative comments across insect species: While I appreciate the introductory comments that situate *Megaselia* extra-embryonic development relative to that in other insects, I am unclear on certain assumptions. Why assume that the change to squamous cell shape is either synchronous or cell autonomous (Introduction, second paragraph, subsection “De-coupling of serosa and yolk sac is preceding serosa tissue spreading”, first paragraph; cf. passive expansion of dorsal cells in the cited Rauzi et al., 2015 paper)? What are the supposed "critical elements of ancestral extraembryonic tissue spreading"? While there is gradual serosal spreading over the yolk in the very large eggs of Orthoptera, this is remote from the region where the embryo is enclosed. Elsewhere, the development of free edges at the periphery of the early serosa is – across the insects – an unusual feature that is only known in restricted, derived holometabolous lineages.

5) Language clarifications: Please consider revising the language used for dynamic locations in different embryos/ species during morphogenesis (e.g., Discussion subsections “*M. abdita* forms an open lateral amnion” and “Free serosa spreading in *M. abdita* requires decoupling from yolk sac”). I believe I can follow what the authors mean, but phrasing such as "until the posterior of the extending germband reached about the middle of the embryo" sounds like a spatial logic puzzle. (Here, should "embryo" in fact be "about 50% egg length along the A-P axis" or simply "egg"?)

6) Please state either in the Materials and methods or in the Results what temperature embryos were incubated at during live imaging, so that the reader can appropriately interpret the many values given in terms of absolute time in minutes.

7) The final Discussion section on the mechanical aspects of the evolutionary origin of the *Drosophila* amnioserosa is succinct and appealing, but why assume a "gradual" loss of tissue decoupling?

*Reviewer #3:*

The authors present a massively improved manuscript. First of all there is now a clear story line that moves us from descriptive anatomy powered by light sheet microscopy towards mechanistic understanding how tissue-tissue interaction affect developmental processes and may contribute to the evolution of embryo morphology. The authors took my (and other reviewer's advice) and prepared outstanding schematics that guide the reader in the tissue topology of this new developmental system. The first three chapters are descriptive and one could argue that the in depth discussion of amnion cell behaviour disrupts the flow of the narrative. However, in a journal such as *eLife* there is space and one cannot object to the high-level of scholarship shown in this part of the paper. The story picks up when the authors detect a pause in serosa spreading. They then ask whether it could be due to interaction with yolk and use a great genetic/embryological trick to differentially label the yolk and serosa. They detect strong correlation in pulsatile behaviour of the two tissue layers suggesting mechanical coupling. This coupling ceases in later stages. Remarkably an MM1 homolog expressed in yolk seems to somehow mediate this coupling. When it is knocked down the coupling persists, causing delays in serosa spreading. Remarkably, the mutant *Megaselia* embryos, at least for a little while, look similar to *Drosophila* where serosa spreading over the embryo does not occur at all. This alone justifies the evolutionary speculation presented at the end of the paper.

One could pick on many details and/or suggest more in depth dissection of the molecular mechanism, but I don't see a reason for that. The story is clear, it has a discovery in it regarding the impact of tissue interaction on morphological evolution, while leveraging state of the art technologies and creative data analysis techniques. I personally would like to see it published as is.

---

## [Author Response]

All three reviewers agree that the manuscript addresses the very interesting question about the nature of morphogenetic transitions accompanying evolution, using the comparative analysis of the development of insect extraembryonic membranes as a paradigm and found the conclusions presented in the manuscript both intriguing and thought provoking. Specifically, the suggestion that the morphogenetic transitions arose, not from a change in the transcriptional differentiation programs that affect fate, but rather from a genetic change that modulates extra embryonic tissue interactions is very interesting. However all reviewers had substantial concerns about the strength of the evidence for some of the conclusions drawn and found the arguments hard to follow. I summarise the major concerns below.i) The premise and the evidence to suggest that the evolutionary change accompanying EE morphogenesis is gradual rather than sudden is poor.

We have changed the angle of narration. Following the next point (general point 2) as well as comments of reviewer #2 and #3, we present our results now as a functional study on extraembryonic tissues in *Megaselia abdita*. Speculations on evolutionary implications have been shifted to the last paragraph of the Discussion.

ii) The suggestion that the modification of inter-tissue (inter EE membrane) interactions contributed to the differences in morphogenesis is the most interesting conclusion of the manuscript. However, the conclusion is neither substantiated by the visualization of these interactions/ topologies, nor validated functionally. Transverse sections from fluorescently stained embryos or EM sections and the analysis of ECM components in wildtype and mutant development will help strengthen this conclusion.

We highly appreciate the suggestion of experiments to visualize and quantify tissue-tissue interactions. Due to technical challenges of fixation (high pressure freezing and freeze substitution, followed by embedding - which seems particularly challenging for tissue interfaces; Yannick Schwab, personal communication, July 2018), we have not been able to visualize tissue interactions in EM sections. We also are not aware of cross-specific antibodies that could visualize ECM components in *M. abdita*. However, we succeeded in implementing the approach outlined by Goodwin at al., 2016, in which the yolk sac membrane was fluorescently labeled by a Basigin-FP fusion. Correlation analyses of movements in yolk sac and extraembryonic tissue (labeled by Lifeact-FP) then allowed us to directly address tissue-tissue interaction in wildtype and RNAi conditions. Results from these experiments (presented in Figure 4 and 5) allowed us to strengthen functional statements that were rather speculative in the first version.

iii) The analysis of cytoskeletal, cell and tissue dynamics, including actin based protrusions, cell shape changes, cell spreading and what the authors describe as 'flipping over" is not of sufficiently high resolution and based on a small sample size.

We thank the reviewers for pointing out that we were over-interpreting our data in some instances. This was not intended and also not required for the conclusions we wanted to draw. Accordingly, we toned down corresponding statements, in particular with respect to actin dynamics in the amnion. Where necessary (the interaction of serosa and yolk sac), SPIM data was substituted by confocal data to provide the required spatiotemporal resolution (see in particular Figures 4 and 5).

iv) The comparative analysis between the three modes of development (ancestral, intermediate and derived) can be very valuable but it is not clear from the data presented whether the developmental stages being compared are appropriate.

Following the reviewer’s suggestion (general point 1), we have limited comparisons between species in the Results section. To allow for better comparison across species, in particular of *M. abdita* and *D. melanogaster,* references to established staging were added.

v) The manuscript is, as it is presented, very descriptive and hard to understand. It could be improved in clarity through the use of schematic representations of the comparative anatomy/ topology of the EE membranes and of timelines indicating the major events accompanying the morphogenesis of the EE membranes.

To improve readability and rationale, we have aimed to streamline and focus the text. In this process we have moved results that were not immediately required into the supplemental figures. As a result, the manuscript now encompasses three mainly descriptive figures (Figures 1-3), which set the stage for our discovery of serosa/yolk sac interaction, and two mainly functional figures (Figures 4 and 5), in which we study the phenomenon and genetic regulation of serosa/yolk sac interaction. In addition, timeline and schematic representations have been added in the hope that this helps the reader to follow development and folding of various involved epithelia.

vi) Both the images and the videos lack sufficient detail in the legends with respect to what is being shown and will also benefit from better labeling.

Labeling, as well as image and figure legends have been reworked. In particular, we aimed to provide better descriptions for projections, axis labels, error bars, color coding, etc. The legends for videos were completely rewritten; we aimed to describing ongoing processes and are providing technical information. In particular for Figure 2—video 1 and Figure 2—video 2, we have added an extra frame in which we indicate cutting plane and highlight structures of relevance. We hope this will help reading the figures and identifying ongoing processes in the videos.

vii) As outlined by reviewer 2 in the detailed comments below, the controls used in the genetic manipulation experiments (RNAi, CRISPR/Cas) should be clearly documented, and the key defining features of wild type serosal and amniotic cellular properties starting at blastoderm stages (Figure 2) must be derived from more than a single biological specimen.

Genetic manipulation experiments have been controlled for artifacts due to injection and possible off-target effects. To control for artifacts due to injection, we have performed injections with recombinant Lifeact-GFP (wildtype control), as outlined e.g. in Figure 4A-F. To control for putative off-target effects in genetic manipulation experiments, we used RNAi and CRISPR as two mechanistically independent approaches for gene knockdown. Key defining features of *M. abdita* extraembryonic development (such as the pause in serosa spreading, the sequential disjunction of the serosa, or tissue-tissue contact with the yolk) were confirmed in at least three time-lapse recordings, in fixed specimen (such as apical area increase of extraembryonic cells), or independently by previously published and cited analyses (such as width of the serosa and amnion anlage in the blastoderm).

The detailed reviews of all three reviewers are pasted below.Reviewer 1:[…] i) The dynamic attachment and/or detachment between the extraembryonic membranes in the wildtype is not easily discerned in the videos or images presented. Could the authors label one but not the other layer and use nuclear markers to enable the clear visualization of both layers? Alternatively, can the authors show images and snapshots from videos in which the outlines of cells of the two layers drawn out (after one layer has been digitally peeled off) and projected to show how these associations (and shapes) are dynamically changing? Also, orthogonal or real transverse sections of the anterior, middle and posterior regions from Lifeact/phalloidin and DAPI- labeled embryos at different stages might help illuminate the topologies of interacting interfaces better.

We agree with the reviewer that tissue-tissue dynamics between the extraembryonic membranes in the wildtype were not easily discerned in non-marked snapshots of the videos. We have now marked snapshots (Figure 2 and 3) to outline amnion and serosa tissue topology. In addition, we have substantiated coupling and decoupling between yolk sac and serosa in cross correlation analyses after using different reporters for actin in the serosa and a transmembrane anchor for the yolk sac. We hope that both approaches help to address the raised concerns.

ii) The stages at which cell and tissue morphogenesis between Drosophila and Megaselia are being compared is not very clear in the data represented in Figure 6. Could the authors include a Materials and methods section that describes staging? This is particularly important as most of the dynamic analysis of the Drosophila amnioserosa has been done at and after the onset of germband retraction.

We apologize for the omission of comparable staging information, in particular since this has been introduced for *D. melanogaster* and *M. abdita* by work of Wotton et al., 2014. The revised text now includes references to these previously described stages.

iii) The dynamic interactions between the extraembryonic membranes and the yolk sac, both in wildtype and in the mmp mutant embryos is not clear. Demonstrating this will be particularly interesting in light of the fact that the yolk cell has been shown to be necessary for both germband retraction and dorsal closure in Drosophila through its interactions and close apposition with the amnioserosa. Here, integrin laminin and basigin have been shown to mediate their interactions. Can the authors examine the deposition of laminin at different stages and their alteration in mmp mutants? Alternatively, and especially as mmp mutants do not seem to affect the morphogenetic process, could the authors examine the distribution and effects of the loss of one of the three components mentioned above?

We agree with the reviewer that the functional analyses of integrin, laminin, and basigin will provide deeper insights into the molecular regulation of tissue-tissue contacts between yolk sac and serosa reported here. As specifically suggested, we have performed preliminary experiments to examine the *Mab-bsg* RNAi phenotype. Our results suggest that knockdown of *basigin* does have an effect on serosa development, but we currently cannot conclusively say whether it affects serosa maintenance, spreading, or adhesion to the underlying yolk sac. We believe that a detailed analysis of the outlined genes will be very interesting, but the results do not seem to be quite straight forward in their interpretation. We feel that they would not add to the statements we currently wish to make, and we would therefore like to propose that these analyses are beyond the scope of this manuscript.

iv) Schematic cartoons and timelines describing the major events accompanying the early morphogenesis of the extra embryonic membranes will aid a better understanding of the manuscript which is otherwise very descriptive.

Timeline and schematic representations have been added.

Reviewer 2:[…] Major concerns in detail:I found the polemical framework of saltatory versus gradual evolutionary change to be rather muddled and of questionable relevance. Anecdotal transition state fossil evidence such as for Archaeopteryx seems to be a weak basis to assume that the evolution of epithelial morphogenesis must occur gradually. Developmental characters such as the presence or absence of abdominal appendages in arthropods provide a clear example of specific genetic changes having fairly abrupt, nearly binary morphological outcomes. Even when I try to disentangle the authors' ambiguous usages of the term gradual, which seems to either refer to evolutionary change or to developmental temporal progression (both meanings seem relevant in the middle of the penultimate Discussion section paragraph), I do not see a reason to posit its relevance here. For example, unless inter-embryo phenotypic variation is seen for the RNAi and CRISPR mmp1 experiments, it appears that the serosa either spreads or does not, but no intermediate levels of spreading are shown. Also, I would caution the authors that developmental genetics investigations in extant species does not really provide a forum for "experimental evolution" and that development in Megaselia does not constitute an ancestral or transition state.

We agree with the reviewer. It was not our intention to make muddled statements. To address this and similar concerns raised by the other two reviewers, we have substantially changed the narrative and have moved statements on possible evolutionary implications to the Discussion.

The significance of the findings is not very strong. In the Introduction the reader is informed that the "absence of a ventrally closed amnion" is "characteristic for this intermediate mode" seen in Megaselia. Why, then, is it "surprising" in the third Results section to find a ventrally open amnion? I was, however, surprised that the authors do not directly address how their data support a major topographical revision of the Megaselia amnion compared to the many cited publications that presented this tissue as being dorsally closed, over the yolk. Rather, a very thin tissue region with a morphologically distinct cell type that forms a lateral ring around the embryo seems to be very novel – not something that can readily be homologized with the ventrally-closing amnion of Tribolium. In general, I would ask the authors to consider the bigger picture context of mature tissue physiological function in assessing potentially homologous characters of the EE tissues, such as serosal cuticle secretion (or other protective and physiological roles performed by a completely enclosing serosal sack) or formation of a sealed, fluid-filled amniotic cavity. In several places in the manuscript, I also found interspecies comparisons specifically for morphogenetic features to be rather casual and vague.

Following the reviewer’s suggestion, we aimed to better embed our description of the lateral amnion into the available literature. The relevant section now reads: “Previous analyses of *M. abdita* amnion development have led to conflicting hypotheses regarding its position and topology. […] Thus following development of the presumptive amnion through consecutive stages of development, our results suggested that the *M. abdita* amnion consisted of a lateral tissue, which was essentially one cell wide, and a cap at the posterior end of the germband that closed over the ventral side of the embryo (which, because of the extended germband, faced the dorsal side of the egg, Figure 2A-D).”

We also thank the reviewer for pointing out the significance of the lateral amnion and the lack of a sealed, fluid-filled amniotic cavity, which we aimed to place into context in the Discussion. The relevant section reads: “The implications of such a reduced lateral amnion on embryonic fly development in general are difficult to assess, in part because the precise function of the ventral amnion in insects is still unknown (Panfilio, 2008). […] Alternatively, a ventral amnion may be formed if the area that had to be covered was smaller, e.g. if the germband was more condensed or even participated in the initial ventral cavity as reported for *T. castaneum* (Benton, 2018).”

Beyond these issues of interpretation, the authors posit a number of intriguing aspects regarding tissue interactions, but they lack the resolution and the functional testing that would be required to support such suppositions. This includes descriptive, correlational observations for the claims that (a) serosal spreading is autonomous and active (so what's distorting the serosa in the mmp1 knockdowns?), (b) there is a yolk membrane that experiences differential adhesion to the serosa over time (not all insects have a yolk membrane and it can vary across species in structure), (c) that oscillations within the serosa are mechanically relevant for serosal detachment. Drug treatments or laser cutting could address some of these mechanical predictions, while high resolution visualization in fixed specimens (TEM and/or immunohistochemistry) would clarify ECM structure and the nature of potential cellular junctions at tissue interfaces.

As outlined above: We very much appreciate the suggestion to visualize and quantify tissue-tissue interactions. We hope that our in vivo approach and cross correlation analyses of movements in serosa and yolk sac can address the raised concerns. Results from these experiments are now presented in Figure 4 and 5.

The sample sizes are alarmingly small. I fully appreciate the big data and computational demands involved in light sheet microscopy work, but some key observations are based on analyses of only a handful of cells from a single biological specimen. Nowhere in the figures, videos, or analyses did I see use of the histone-labeled, lower magnification recordings, which cannot visualize cell outlines and already curtails the effective dataset by 25%. Furthermore, not all recordings span the same developmental stages: the key defining features of wild type serosal and amniotic cellular properties starting at blastoderm stages (Figure 2) therefore derive from a single biological specimen, in which only 23 cells were examined quantitatively (it is unclear whether this value is even per tissue type or in total across three distinct tissues). Even for developmental stages documented in multiple recordings, there is only a sample size of 2 – or in one case 3 – individuals, and most quantitative plots are for fewer than ten cells (unclear if multiple specimens were used in such analyses, e.g., Figures3B, 6D, 6I, 7H). For example, is the sequence of serosal detachment (anterior, posterior, then lateral) consistent across specimens, and what is the error or variability in calculations of tissue area increase over time?

As partly outlined previously, key defining features of *M. abdita* extraembryonic development (such as the pause in serosa spreading, the sequential disjunction of the serosa, or tissue-tissue contact with the yolk) were confirmed in at least three time-lapse recordings, in fixed specimen (such as apical area increase of extraembryonic cells), or independently by previously published and cited analyses (such as width of the serosa and amnion anlage in the blastoderm). To address concerns on effects of SPIM imaging, we have added additional observations on embryonic development, which we compare we descriptions in the literature. Based on selected hallmarks, we find embryonic development in *M. abdita* very robust. We would therefore argue that, as long as key features are independently confirmed (see above), presentation of a representative time-lapse recording is sufficient for the statements we aim to make in the Results section.

The use of an F-Actin fluorescent label to characterize cell shapes provides only limited resolution, and I cannot clearly see a number of features described by the authors. In actively remodeling tissues, F-actin is not the most reliable indicator of cell shape. Since labeling was achieved via injection of an existing construct, why was a membrane label, such as Gap43-YFP not used additionally or instead? Even in frame-by-frame inspection of the manuscript's videos, I find it very difficult to fully discern individual cell outlines at all time points or to see the nature of the tissue organization dorsal to the amnion (in peeled off serosal renderings). The cut-away volumetric renderings are helpful, but it does not appear that F-Actin clearly or fully label the basolateral domains of cells, which is critical for ascertaining cell shape and volume. The use of a ubiquitous label also precludes rigorous tissue discrimination and cannot distinguish apical-basal polarity, which seems crucial, for example, in confirming the authors' claims that amnion cells "flip over". Moreover, F-Actin does not label the yolk sac.

We agree with the author that F-actin is not the ideal indicator of cell shape. We would like to argue, though, that for the purposes of crude discrimination between serosa, amnion, and ectoderm cells, F-actin is sufficient and provides additional information, e.g. the actin protrusions visible in the late amnion during germband retraction. We furthermore agree with the reviewer, that by using the term “flipping over” we made a cell-biological statement that may require additional data to be substantiated. However, our conclusions do not depend on such a statement, and we feel that a detailed analysis would be beyond the scope and main message of this manuscript. We therefore decided to not use the term “flipping over” any longer. To label the yolk sac, we are now using a Basigin-FP reporter.

These experimental design limitations seem to carry over into the analyses in several places. There are several imprecise renderings of image data that impair clarity and do not create a strong impression of robust analyses for a study that is largely based on manual tracking. For example, there is a discontinuity in labeling (and ability to identify?) amnion cells in Figure 1F-I (why is there a gap posterior to cell #2?). Furthermore, tinted regions to indicate cells and tissues seem somewhat arbitrary or imprecise with respect to the F-actin signal (Figures3C and 4D, and Video 6). In Video 6, the tinting to indicate the entire serosal tissue is markedly inconsistent between consecutive time points regarding which cells are included, and the tinted region often does not respect cell boundaries. This seems to be in contrast to consistent tissue edge shapes over time as shown in Figure 5B. Furthermore, the patchy, limited scope of wild type Mab-mmp1 expression seems to contradict a role in a membrane layer at the yolk's surface. Also, can the authors reconcile the lengths of protrusions observed by live imaging (which are very hard to see) with that documented by expression of Mab-egr transcript (compare Figure 3D-E with 3G')?

Presumably due to injection of recombinant Lifeact-FP, we have observed higher background in our recordings than what has been reported for transgenic reporters in e.g. *Drosophila melanogaster*. Our tracking of individual cells has been limited by these technical challenges, but we believe that they do not impact on the conclusions we are drawing. In the particular case of putative amnion cell tracking, the key result are not the tracks per se. The conclusion we want to draw is that the putative amnion (which consists of large, non-dividing cells that remain behind when the serosa detaches and grows over the embryo) stems from a single line of cells next to the serosa anlage in the blastoderm, and we believe that the presented results in context of previous and cited findings are sufficient to draw this conclusion.

Imprecise marking of the serosa, which was apparent in the overlay of fluorescent signal and tinted region, stemmed from an artifact that was introduced by imprecise 3D rendering. The rendering has been repeated and tinted region and outline of serosa now overlap.

We would not consider *Mab*-Mmp1 expression to be patchy but rather to reflect enrichment of transcripts around the yolk nuclei. Since the protein is presumably secreted into the space between yolk sac and amnion, we would argue that such an enrichment does not contradict a role in a membrane layer at the yolk's surface.

Specific methodological concerns:How many RNAi and CRISPR/Cas experiments were performed (number of instances of injection)? Were multiple, non-overlapping dsRNA fragments used for Mab-mmp1 to ensure specificity of knockdown (avoid off-target gene knockdown effects)? Can the authors comment on the efficacy of CRISPR/Cas with respect to biallelic editing or the degree of mosaicism? It may be that failure of serosal spreading is a generic phenotypic outcome when embryos are unhappy: what negative control (i.e., targeting a completely different gene) was performed?

Genetic manipulation experiments have been controlled for putative off-target effects by using two presumably fully independent approaches to knock down gene activity. With respect to CRISPR, we cannot make statements on biallelic editing or the degree of mosaicism. However, we would argue that this may not be necessary. The key statement we want to make is that reduction of *mmp1* activity reduces/delays de-coupling of serosa and yolk sac. We feel that by using two different approaches are sufficient to make this statement, even though each approach by itself likely reduces and does not fully eliminate *mmp1* activity. Similarly, we would argue that failure of serosa spreading is not a generic phenotypic outcome when embryos are unhappy. All our time lapse recorded embryos were injected once, embryos analyzed for cross-correlation were injected even twice – and still they showed what we would argue are very specific differences in serosa behavior between otherwise wildtype or *mmp1* RNAi embryos.

Given that oscillations in apical area are the key distinguishing feature between the Megaselia amnion and either the Megaselia serosa or Drosophila amnioserosa, please strengthen the quantitative presentation of this. What is the periodicity, duration of the oscillatory phase, developmental stage of onset and cessation, and level of inter-embryo variability? Why were other quantified features such as apical area and apical circularity not also applied in characterizing the amnioserosa?

After we substituted previous SPIM analyses with confocal data at higher spatiotemporal resolution, we no longer make a statement about oscillation. The corresponding section has been removed from the manuscript.

As injection and fluorescent live imaging can interfere with embryogenesis, what controls did the authors take to ensure that these treatments did not impair healthy development?

We thank the reviewer pointing out previously missing information. To control that our treatments did not impair healthy development, we compared staging with previously published descriptions of embryonic development (Wotton et al., 2014.) as well as our own observations. This information was previously missing, and we thank the reviewer for pointing this out. The relevant section in the Results section now reads: “Next, we tested whether overall development was affected by long-term imaging of injected *M. abdita* embryos. […] In our quantified SPIM recordings we found the same timing of events, suggesting that development of injected *M. abdita* embryos was not notably affected by long-term SPIM imaging.”

Insect development is temperature dependent, yet many events are presented in terms of absolute time in minutes. Please take care to note the temperatures for incubated eggs (such as prior to in situ hybridization) and during live imaging recordings, and to relate these clearly to developmental stages.

We apologize for the omission of this information. The relevant information has been added (e.g. the time of *Mab-ddc* staining in Figure 5 and Figure 5—figure supplement 1). Furthermore, we have introduced staging information and developmental landmarks as reference points to allow for better comparison between species.

As detailed as the Materials and methods section is, I do not find the light sheet and point cloud sections sufficient to be reproducible. Actions such as "smoothening" and "eroding" are only informative in light of the operating system/ software/ custom scripting/ other, which needs to be cited and or made available (e.g., via GitHub).

We have aimed to better indicate Matlab functions that were used in processing the data. Custom functions are indicated by capitalized first letter and are available via GitHub.

Presentation concerns:The entire paper would strongly benefit from a substantially strengthened explanatory framework for Megaselia development, with: schematic representations of sampled regions over the egg circumference and of specific cell shapes and tissue boundary organizations; a clear statement of developmental staging and landmarks; and explicit definitions of quantified features. For example, when exactly does serosal spreading initiate and complete relative to germband extension and retraction? How exactly was apical circularity calculated and what is the "ideal circular apex"?

With the suggested change of narrative, we also strengthened the framework of *M. abdita* development. We aimed to provide better schematic representations of sampled regions and better definition of quantifications. Key features of serosa development are now placed in the context of germband extension (Figure 3D), and apex circularity is defined in the legend of Figure 1—figure supplement 1. The relevant section reads: “Apical circularity (c) was defined as c=1 for a perfect circle and c < 1 for angular shapes with c = 4 π area/perimeter^2^ (Thomas and Wieschaus, 2004).”

Across both figures and videos I found serosal separation from the blastoderm impossible to see clearly. How was this determined kinematically, and why is there ambiguity about this in the oldest stage analyzed for Figure 1? Similarly, given the 4D reconstructions from SPIM data, why is no ventral view to illustrate serosal closure provided?

To kinematically determine the serosa disjunction, we took advantage of an enrichment in fluorescent signal of our actin reporter at the interface of serosa and amnion. This is now described in Figure 3—figure supplement 1. The relevant section reads: “Serosa disjunction was determined kinematically by visually following actin enrichment between serosa and amnion. […] This process was best resolved by repeatedly playing recordings forward and backwards in time.”

Color legends and axis labels are missing in several cases and/or are not defined in the figure legend. For example, what is the primary y-axis in Figures 1E-G, 3B, 5D-E, and what does "GBE [%]" mean as a y-axis label in 6E, 6J, 7I (given that 100% GBE is never achieved)? What is the secondary y-axis label in 2J and 3B? Please improve the text resolution for in-figure labels of samples sizes, and please explicitly indicate how many biological specimens and tissue types are included in any given n. For the very limited sample sizes, please provide (schematic) insets to show where along the embryo surface those cells are located. Please define colors such as for the heat maps in 5D, 6C, 6H, 7F (is darker older or younger?), and the various use of pinky-purple in plots compared to blue-turquoise for cell traces (6D, 6I). What are the heat maps on certain plots (5C, 6B, 6G, 7G)?

To improve image legends, we revised all figures and figure legends provided in the manuscript. We included detailed descriptions of all axes and error bars in either the figure itself or in the figure legend. Readability of axis labels were improved and where missing, heat map legends were included and described in the corresponding figure legends. To provide orientation within the organismal context, schematic representation of different embryonic stages were included. Descriptions of close-ups and embryo orientation were revised and now always include orientation in anterior posterior and dorsal ventral. Colors and color codes were completely renewed and are explained in either the figure or the corresponding figure legend.

There appears to be a displaced box for the ROI in Figure 1A: the actual cells shown at high magnification (1B-C) are just posterior to the position of the box in 1A.

The indicated ROIs and close-ups of former Figure 1 (now Figure 1—figure supplement 1) have been adjusted and are now indicating the correct position of the ROI on the surface of the embryo.

What are the error bars in Figures 2J and 3B?

We completely revised the figure legends and they now include the definition of the error bars, which are defined as standard error of mean (Figures 2, 4 and 5).

Presumably the first time point in Figure 3D-E is magenta, not cyan.

Previous Figure 3 was completely reconstructed and the panels D-E were removed from the manuscript.

Presumably the images in Figure 3D-E are shown in dorsal-flattened aspect with anterior up and centered on the dorsal midline (constituting a 90-degree rotation relative to the images in 3F-G), but I should not have to guess these things!

To improve readability of the individual figures and panels, figures and figure legends were completely revised. Former Figure 3 was completely reconstructed and the panels D-E were removed from the manuscript. Wherever needed, the orientation of close-ups and whole specimens were included in the figure or legend.

There is no figure legend for Figure 6 panel E.

Figure assembly of all previous figures were revised and panel 6E is not shown anymore. All figure legends were rewritten and are now describing all panels in the corresponding figure.

Please take care to distinguish an entire egg/ biological specimen from embryonic tissue proper. Using "embryo" to refer to both is very confusing, particularly for blastoderm stage events.

Following the reviewer’s suggestion, we use the term “embryo proper” or “ectoderm” where we want to differentiate between development in extraembryonic epithelia and embryonic epithelia.

Please be consistent in nomenclature for species (M. abdita or Megaselia) and always italicize.

We thank the reviewer for pointing out these inconsistencies. We now refer to *Megaselia abdita* as *M. abdita* throughout the text.

I have no stylistic objections to the use of more informal language so long as meaning is clear. But why not use "columnar" and "squamous" as appropriate to define cell shape?

The paragraphs describing wildtype extra-embryonic tissue development was completely revised and now includes cell shape and tissue descriptions as suggested.

Several brief sections were not clearly introduced and their relevance and significance were not apparent, such as the exploration of the anterior amnion and the very detailed analysis of zebrafish EVL expansion.

We have aimed to provide better introductions and rationale for the presented experiments. The analysis of zebrafish EVL expansion has been removed as suggested.

Reviewer 3:[…] Introduction is well written and clear. However, the model system used is relatively unknown and a schematic of the anatomy and current knowledge of the dynamic tissue remodelling events would help the reader orient in the subject. Such schematic should then be used consistently in most of the following figures that are hard to follow without conceptual visual aid.

We have provided schematic overviews to summarize current knowledge and our new results. Once assembled, we realized the value of these overviews and would like to thank all reviewers for this suggestion.

In their first experiment, the authors quantified cell shape in fixed stained Megaselia embryos at several stages of development. The data are clear when it comes to apical cross section of the cells. From the quality of the images, even in the zoomed-in versions, it is less clear how accurate could the automated segmentation of cell shapes be in the axial direction (along the apical basal axis of the cells) where the resolution must be necessarily significantly lower. Could the authors show in supplements such cross sections and comment on the reliability of their Ilastik segmentation pipeline on this data? The PCA analysis is summarized in many plots that are largely redundant.

As the reviewer pointed out the segmentation was challenging in some areas, which were then corrected manually. The relevant section now reads: “In the case of mis-segmentation by the automatic algorithm due to lower resolution, cell outlines were corrected manually.” Following the reviewer’s suggestion, the PCA analysis was removed from the manuscript.

Tracking experiments in SPIM recordings show convincingly that serosa cells are able to spread over adjacent cell layers and that these cells can be mapped onto the region of the blastoderm from which serosa originates (based on zen expression – Figure 2E could use some context). The cells next to the serosa cells are identified as amnion based on their size and cell division status and the fact that they are overgrown by serosa and eventually separate from it. In the absence of clear genetic evidence for the mapping of these cells to amnion, it would be prudent to refer to them as "putative amnion" rather than "amnion" as the authors chose to do in the rest of the paper. It is unclear how the last paragraph of this section tests the robustness of tracking. This confusion stems perhaps from the lack of clear conceptual explanation of what the serosa amnion separation means and what behaviour the amnion cells are expected to show.

As suggested, we adopted the term “putative amnion”. Due to data quality, anterior amnion tracking could not be confirmed in more than one recording and was therefore removed from the manuscript.

In the next set of analyses of the SPIM movie the authors follow the ultimate fate of the putative amnion cells. The section is rather confusing. On the one hand, authors are surprised that the lateral amnion remains ventral and yet they seem to expect that the amnion will close ventrally (first paragraph). The Mab-eiger staining reveals, presumably viewed from the dorsal side, that amnion does not close dorsally and that the cells have protrusions. I appreciate the difficulty of visualizing the live in toto imaging, however the statements about cells "flipping over" are completely unclear from the data. Also, the evidence for the filopodia like protrusions is not convincing from the live imaging data. Given these difficulties in understanding the text and the data, the strong evolutionary conclusion about "gradual transition in the evolution from the ancestral ventral amnion" cannot be easily accepted. I propose a thorough rewrite of this section and to enhance the data with conceptual drawings for further clarity.

As suggested by the reviewer, and also in light of the changed narrative, the section has been completely rewritten and we no longer speculate about a "gradual transition in the evolution from the ancestral ventral amnion".

To substantiate the "flipping", is it possible to confirm the apical-basal side of these cells with an apical polarity marker?

We realized that by using “flipping over” we made a cell-biological statement that we actually did not intend to make. We therefore decided to tone this down and now describe development of the putative amnion as follows: “During germband extension and up until the onset of germband retraction, cells of the presumptive lateral amnion had a rather smooth outline and appeared to be folded over the ectoderm; […] the serosa first separated from the presumptive amnion and then appeared to drag it slightly over the adjacent ectoderm. As a result, amnion cells seemed to be turned with their lateral side towards the basal membrane of the serosa”. And, specifically to the question: we imagine that it is in principle possible to substantiate “flipping” by observing an apical polarity marker. Unlike in *D. melanogaster*, however, such markers are currently not yet established for *M. abdita*. Because we believe that a more detailed analysis would not add significantly to the main findings reported here, we would like to propose that analyses with an apical polarity marker are beyond the scope of this manuscript.

The authors next describe a supracellular actin cable between serosa and amnion. While I believe that such a cable exists, the pictures in Figure 4 do not show it very clearly. The lines mostly obscure the presumed accumulation of actin. Please show the panels with and without the lines. At the end of the second paragraph, authors state that "the serosa continues to spread laterally and eventually broke free, leaving behind a single row of amnion cells […] the actin cable was observed in the expanding serosa but not the remaining amnion." This seems important and would be worth substantiating with some panels in the figure.

We thank the reviewer for pointing out that we used obscuring labeling to highlight the actin cable. To illustrate this phenomenon more clearly, we are now providing improved labeling in Figure 2—figure supplement, Video 1, and included a new supplemental figure that describes how we have been using the actin cable in delimiting the extension of the serosa (Figure 3—figure supplement 1).

Next the authors discovered pulsations in the extraembryonic tissue. They claim that pulsations were observed only in cells of the serosa that are not participating in the cable. To me the data in Figure 5D, E show that both border and non-border cells pulsate, while border cells perhaps less. A statistical treatment of the data is required. Moreover, authors speculate that the pulsation decouples the serosa from the yolk sac. Several questions arise. Apparently, the increased cell oscillations coincide with the pulsation of the yolk. Are these pulsations in the yolk cells? Please show the yolk pulsations data. What indicates that the pulsations decouple the serosa from the yolk?

In this version of the manuscript we no longer refer to oscillation that we reported in SPIM recordings of *M. abdita* wildtype development. We still see these oscillations, and they remain visible in the videos we provide in the supplemental information. However, we obtained complementing results of yolk sac dynamics by confocal microscopy with Basigin-GFP as fluorescent marker (see also general point 2, above), which indicate that oscillation in the yolk sac is not constant over time. Analysis of these oscillations in the yolk sac are ongoing. However, they will require additional time, and we feel that they were not necessary to address the interaction between serosa and yolk sac.

The authors next turn to genetic perturbation experiments. These are non-trivial experiments in Megaselia, in particular when coupled with complex SPIM imaging. In Zen RNAi serosa is transformed into amnion and accordingly there is no spreading. Authors report lack of pulsatile behaviour that they interpret as amnion-like. In Figure 6D I lack quantitative comparison of the extent of pulsatile behaviour to either serosa or amnion wildtype cells. In Drosophila, amnioserosa cells do not spread but pulsate extensively. Authors confirm this with their own in toto recordings. I am not sure what can one conclude from comparing the germband retraction dynamics of wildtype Drosophila and Zen-RNAi Megaselia. Therefore, the evolutionary statement at the end of the paragraph remains highly speculative.

We agree with the reviewer that the analysis of *zen* RNAi embryos does not add substantially to our analysis of tissue-tissue coupling between serosa and yolk sac. We have therefore decided to remove the results from the manuscript.

I find the final experiment of the paper most impressive. Knocking down extracellular adhesion molecule mm1 in Megaselia prevents serosa spreading, the embryos strikingly resemble Drosophila and yet development is unaffected. Analysis of SPIM recordings shows pulsations however it is claimed that it is dampened. Again, this claim needs to be better substantiated. The connection between serosa cells and yolk, proposed two sections ago, gains some traction here and in fact should be presented together.

We very much appreciate the encouraging comment by the reviewer and hope the our new data and cross-correlation analyses of movements in serosa and yolk sac provide the necessary support for our conclusions.

Overall, I find the paper thorough, thought provoking and valuable to the scientific community. The overall pitch towards the fundamental question of the gradual evolution of developmental mechanism seems a little bit exaggerated. It belongs to the Discussion and should not drive the narrative of the paper. The data simply do not (and cannot) support broad statements such as 'amnioserosa in Drosophila evolved by loss of serosa spreading without disrupting the developmental programs of serosa and amnion'. This does not mean that the authors cannot speculate in that direction. A clear conceptual model figure at the end would highlight the exciting evolutionary implications of this work.

As outlined above, we have adjusted the overall pitch of the manuscript and moved evolutionary speculations to the Discussion.

[Editors' note: further revisions were requested prior to acceptance, as described below.]

Reviewer #1:[…] i) How does "disjunction" correlate (temporally) with serosal spreading? Could this explain the differences in serosal dynamics between the serosa and the amnioserosa (composite)? What does "disjunction" involve: loss of adhesion between the amnion and serosa? Does Mmp also affect this?

We thank reviewer #1 (and reviewer #2) for pointing out that the revised manuscript still lacked a concise definition of “disjunction” of serosa from amnion, and an explanation of how we believe it relates as event to the “decoupling” of serosa and yolk sac.

In essence, we use the term “disjunction” to describe the loss of adhesion between amnion and future serosa leading edge. We have added this missing information in our description of amnion behavior, where the corresponding paragraph now reads: “To understand how amnion cells behaved during and after their separation from the serosa, we computed donut-like sections of the developing embryo that allowed us to observe embryo surface and transverse section in 3D renderings (Figure 2G; Figure 2—video 1) […]Following this disjunction, the presumptive amnion cells seemed to be turned with their lateral side towards the basal membrane of the serosa.”

While it is in principle possible that differences in tissue dynamics between serosa and amnioserosa could be related to the ability or inability of the serosa to disjoin from the amnion, we do not believe that this is the case here. According to our understanding, serosa/amnion disjunction is a process that follows serosa/yolk sac decoupling, and we do not have evidence that disjunction is affected by Mmp1 RNAi. To better explain why we think that it is “decoupling” first, followed then by “disjunction”, we added the following sentences to the discussion of free serosa spreading: “Notably, serosa disjunction was a gradual process that was first observed in the anterior and last in its lateral periphery. Regardless of location and timing, disjunction was characterized by a very steady and even expansion of the serosa leading edge without signs of tissue rupture, suggesting that tissue-level decoupling from the yolk sac and cell-level loss of adhesion from the amnion had already occurred earlier.”

ii) Does "decoupling" also imply physical detachment between the yolk and the serosa? While the authors argue for decoupling based on correlating movements, it will be good if they can show both in the transverse sections and schematics of the kind shown in Figures 2G and 2A respectively, the positions of the yolk and serosa when they are decoupled. Is the dorsal most serosa still sitting on the yolk while the lateral serosa is not? Confocal real time movies focusing on the midsaggital plane will also be useful to examine the former.

We do not think that "decoupling" necessarily implies physical detachment between the yolk and the serosa. We rather feel that our results suggest that this is actually not the case: we see a strong reduction of coupling, but we do not see it gone completely during late serosa spreading. Our interpretation is that yolk sac and serosa remain physically associated, but they are free to slide past each other. To better reflect this idea, we have changed the corresponding section as follows: “We found this coupling to be significantly reduced after the second phase of serosa extension had started (Figure 4E,F). Notably, coupling was not completely lost, suggesting that yolk sac and serosa remained physically associated, but loosely enough to slide past each other.”

iii) Are the authors suggesting that the fast phase of serosal closure uses the epidermal cells as a substrate for spreading? What happens to the epidermal cells during the process? Are they moving dorsal ward along with the amnion? How does the fast phase of serosal closure correlate (temporally) to the anteriorward movement of the epidermis or the contact of the serosa with the epidermis?

We have not addressed the substrate of the serosa during its spreading: it may be the epidermis, it may alternatively be the vitelline membrane. We consider it unlikely, however, that serosal closure could be influenced by putative “co-movements” of an ectodermal substrate. This is, because at the time of serosal closure, the ectoderm essentially does not move, since all major gastrulation events (mesoderm internalization, germband extension) are completed. We hope that the suggested timeline, which is provided in response to comment #4, helps to further illustrate this.

iv) A timeline indicating the major events involving the extraembryonic membrane including germband extension and retraction and the morphogenetic transitions in the three tissues including disjunction, decoupling, fast, slow and paused phases of serosal dynamics marked on this timeline would be useful. This is particularly important as the manuscript examines the dynamics of the tissues across different morphogenetic movements (extension and retraction of the germband).

A timeline to summarize the relative order of events has been added as Figure 4—figure supplement 2.

Reviewer #2:[…] 1) A crucial feature of the study is the disjunction of the serosa from the adjacent blastodermal epithelium. This is an essential prerequisite for the serosa to spread over and enclose the embryo, and several analyses in the manuscript distinguish morphogenetic behaviors before and after disjunction. How is disjunction ascertained? Nowhere did I find a clear statement of how this was determined. There may also be some confusion on this point, as it is difficult to understand how the serosa could exert displacement drag on the amnion after they have separated (subsection “The amnion in M. abdita develops as a lateral stripe of cells”, last paragraph). Note that qualitative descriptions, such as of the serosa 'breaking free' (subsection “M. abdita serosa spreading proceeds in distinct phases”), carry mechanical implications that are not necessarily justified.

We thank the reviewer for pointing out confusion in this section. The paragraph has been rephrased, and a definition of disjunction has been added. It now reads: “To understand how amnion cells behaved during and after their separation from the serosa, we computed donut-like sections of the developing embryo that allowed us to observe embryo surface and transverse section in 3D renderings (Figure 2G; Figure 2—video 1). […] This orientation was reversed during germband retraction, when amnion cells no longer appeared folded over the adjacent ectoderm but rather over the yolk sac (Figure 2D, I-I’’’; Figure 2—video 2).”

Subsection “M. abdita serosa spreading proceeds in distinct phases” now reads: “In the second phase, the serosa was characterized by free edges at the periphery, cells increased further in their apical area, […]”.

2) Amnion morphogenesis: The supplementary videos and figure associated with main text Figure 2 are excellent, and clearly show the unusual behavior of the lateral amniotic cell row folding over along the dorsal-ventral axis. Does this folding over occur in an anterior-to-posterior progression (the small region shown in the videos suggests this might be the case)? However, at later stages while I can see the dorsally protrusive cell shape changes, I am less convinced that the cells exhibit crawling behavior: (a) the ventral actin cable that delineates the amnion from the epidermis remains quite distinct and straight, suggesting the retention of tissue-level organization, and (b) my impression from the video is that the amnion is rather displaced passively due to other developmental events, while crawling implies active behavior. Please clarify the description of the amnion for these features.

Separation of the serosa from the presumptive amnion occurs in a bidirectional manner, i.e. towards the anterior as well as the posterior. We have added this information in the legend of Figure 2—video 1: “Starting from this local fold-over, the serosa then separates from the adjacent tissue in a bi-directional manner and continues to spread over the embryo proper.”

The notion of “crawling-like” cell behavior in the amnion has been removed. The relevant sections of the Results now read: “with the onset of germband retraction, these cells then developed notable protrusions towards the dorsal midline of the embryo (Figure 2C, D).” and “This orientation was reversed during germband retraction; presumptive amnion cells no longer appeared folded over the adjacent ectoderm but rather over the yolk sac (Figure 2D, I).”

3) The new investigation of serosal-yolk sac interactions is now the centerpiece of the manuscript, emphasized in the revised manuscript title. Given the importance of these experiments, please clarify the following:A) Although perhaps less important than fluorescent reporter localization, the in situ characterization of endogenous Mab-basigin would be an important aspect of characterizing a new gene and labeling tool in Megaselia. Unfortunately, the in situ micrograph is neither convincing nor informative: potential yolk sac signal seems no stronger than potential background signal in the head. In the main text there is a comment about delaying injection of the Basigin label so as to restrict localization. Please reconcile statements on the level of specificity of basigin and of the transgenic label (subsection “De-coupling of serosa and yolk sac is preceding serosa tissue spreading”, second paragraph). For endogenous transcript, sectioned rather than whole mount material would clarify the extent to which expression is restricted to the yolk periphery, even with coarse sectioning by hand.

We have reassessed the in situ staining of Mab-bsg and agree with the reviewer that expression can be detected in the head region. This, however, can also be seen for expression of the *Drosophila* orthologue (link to BDGP in situ homepage below:

http://insitu.fruitfly.org/cgi-bin/ex/report.pl?ftype=10&ftext=FBgn0261822). For this reason we have changed the description as follows: “Analysis of Mab-bsg expression suggested that the gene was expressed in the yolk sac (Figure 4—figure supplement 1).”

Whether or not Mab-basigin is exclusively expressed in the yolk, whether the transcripts are restricted to the yolk periphery, or whether expression is different from *Drosophila* are questions that we feel are outside the limited scope of adapting an established *Drosophila* assay for *Megaselia*. We would like to argue that our aim has been to adapt a *Drosophila* approach for yolk sac membrane labeling in *Megaselia*, with all the limitations of working in a non-standard system. *Drosophila* expression of Basigin-GFP is provided by the protein trap line G289 (Reed et al., 2004; Goodwin et al., 2016). Such a stable transgenic line does not exist in *Megaselia*. To still mimic expression of this line as closely as possible and with least possible side effects in *Megaselia*, we decided to inject mRNA encoding the Basigin-GFP fusion into the yolk sac. Putting aside arguments on expression levels (which are associated with most in vivo reporters, at least, again, in non-standard models), we would argue that this approach is justified as long as endogenous transcripts can be detected in the yolk sac. We have seen yolk sac expression of Mab-bsg consistently in over 40 embryos analyzed at the presented stage (information added to the legend of Figure 4—figure supplement 1), which in our view is sufficient to use the assay as presented. To better explain our rationale, the relevant section in Materials and methods now reads: “Because a protein trap line equivalent to the *D. melanogaster* basigin reporter does not exist in *M. abdita* (Reed et al., 2004), we reasoned that injection of mRNA encoding Mab-bsg-eGFP into the yolk sac could mimic the desired properties of the *D. melanogaster* reporter as closely as possible and with minimal side effects in *M. abdita*. To express Mab-bsg-eGFP specifically in the yolk sac, Mab-bsg-eGFP mRNA was injected after germband extension had started and yolk sac formation is thought to be completed (Rafiqi et al., 2010).”

B) For the material analyzed in Figure 4, where is this in the embryo? Morphologically, it looks as though the epidermis is included in the lower part of the viewing field (Figure 4A). An inset schematic would help for orientation. In the accompanying video, I find it hard to understand the spatial and temporal context of the video on the right. Why are the viewing fields in the video cropped relative to the main text image, and why do they differ between the left and right videos within the file?

An inset schematic has been added as suggested, and the size of the videos has been adjusted to reflect the size that has been used as input for single cell cross correlation analyses. To better reflect this, the following sentence has been added to the video legend: “Videos were cropped to reflect the size that has been used as input for single cell cross correlation analyses”.

C) More generally, spatially and across embryos, how representative is the analysis in Figure 4? It currently appears to report a detailed analysis of only a single, small tissue region in a single specimen. While I note sample sizes within Figure 5, this information could be made more explicit in the main text, along with the spatial information.

Panels 4A-D are representative sample images and an exemplary analysis from a single embryo and cell, and they are meant to qualitatively illustrate the type of data that is summarized for multiple embryos/cells in 4E and 4F. To better reflect this, the legend now reads: “(A-C) Sample images of serosa (visualized with Lifeact-mCherry) and underlying yolk sac (expressing Basigin-eGFP) are shown in average-intensity projections. […] (F) Collective comparison of correlation coefficients for individual cells before and after serosa disjunction, bar indicates the mean.”

We are not sure whether information on sample sizes are required in the main text. Here we have been following common practice in the *Drosophila* field (e.g. Goodwin, et al., 2016, or Levayer and Lecuit, 2013, Dev Cell 26, 162–175.) and provide information on sample sizes for cellular/tissue-level quantifications within the respective panels.

D) The legend to Figure 4 indicates that a t-test was used in assaying serosa-yolk correlation before and after disjunction, but are the data normally distributed such that the assumptions of the t-test are met? (There seems to be a skew towards higher correlation values for the data shown in Figure 4F.)

All data has been tested to fit normal distribution. The information is now added in Materials and methods: “Statistical comparisons of correlation coefficients distribution were performed via Matlab’s implemented 𝜒^2^-square test for normal distribution, Student’s t-test (two sided, unpaired), and hample’s test. All samples fit normal distribution; P-values of Student’s t-test are indicated in the figure legend”.

E) Subsection “De-coupling of serosa and yolk sac is preceding serosa tissue spreading”, last paragraph: why "presumably" in describing potential oscillations? Can these oscillations be linked to biological processes such as serosal spreading or events in the embryo such as segmentation or germband expansion? Given that they serve as a landmark feature in subsequently characterizing the mmp1 loss-of-function data, a brief comment on this developmental feature would be appreciated.

We followed the suggestion of the reviewer and rephrased the sentence as follows: “These analyses detected potential oscillations in membrane behavior along the anterior-to-posterior axis, […]”.

While we share the excitement of the reviewer about a potential link of serosa/yolk sac membrane oscillation with other aspects of embryonic development, we feel it is too early to comment on it as a developmental feature; we simply do not yet understand what is going on. We also believe that for the context of the manuscript and our analysis, it is not necessary to go more into detail. While all of our cross-correlation analyses evaluate movements that may, in part, result from oscillation, the method per se is absolutely ignorant in the type of coinciding movements between serosa and yolk sac and also takes into account other, non-oscillating movements. This is why we do not see potential oscillations as “landmark feature” of our analysis, but would rather consider them as possibly interesting side observation that may be followed up in an independent study.

4) Comparative comments across insect species: While I appreciate the introductory comments that situate Megaselia extra-embryonic development relative to that in other insects, I am unclear on certain assumptions. Why assume that the change to squamous cell shape is either synchronous or cell autonomous (Introduction, second paragraph, subsection “De-coupling of serosa and yolk sac is preceding serosa tissue spreading”, first paragraph; cf. passive expansion of dorsal cells in the cited Rauzi et al., 2015 paper)? What are the supposed "critical elements of ancestral extraembryonic tissue spreading"? While there is gradual serosal spreading over the yolk in the very large eggs of Orthoptera, this is remote from the region where the embryo is enclosed. Elsewhere, the development of free edges at the periphery of the early serosa is – across the insects – an unusual feature that is only known in restricted, derived holometabolous lineages.

We have added the missing reference indicating synchronous and cell autonomous cell shape changes and also toned down the argument. The sentence now reads: “While synchronized and cell-autonomous flattening of cells is likely a contributing component (Pope and Harris, 2008), […]”.

We have rephrased the sentence to better specify what was meant by “ancestral elements”. It now reads: “Because the serosa in *M. abdita* has retained the ancestral ability to expand and envelope the embryo proper, […]”.

Because of results reported by Pope and Harries, we believe it is reasonable to consider a tissue-autonomous program of cell thinning and spreading in the serosa as a possible explanation for serosa behavior. To indicate that this is a possibility that may invite further testing, the relevant sentences now read as follows: “For example, the interruption in serosa expansion could be explained by a temporal pause in a tissue-autonomous program of cell thinning and spreading. However, expression of the tissue-fate determining Hox-3 transcription factor Zen remains high also throughout stages in which tissue expansion paused (Rafiqi et al., 2008), indicating little, if any, change in potential upstream genetic regulation of cell thinning and spreading.”.

5) Language clarifications: Please consider revising the language used for dynamic locations in different embryos/ species during morphogenesis (e.g., Discussion subsections “M. abdita forms an open lateral amnion” and “Free serosa spreading in M. abdita requires decoupling from yolk sac”). I believe I can follow what the authors mean, but phrasing such as "until the posterior of the extending germband reached about the middle of the embryo" sounds like a spatial logic puzzle. (Here, should "embryo" in fact be "about 50% egg length along the A-P axis" or simply "egg"?)

The section in the Discussion has been rephrased to simplify and clarify topological statements. It now reads: “This observation correlates with the size of its blastoderm anlage, which is dorsolaterally narrow and slightly larger only in the posterior blastoderm (Kwan et al., 2016; Rafiqi et al., 2012). […] Further work is required to elucidate genetic mechanisms that distinguish a lateral and a ventral amnion, which could, e.g., develop from a larger anlage, rely on proliferating amnion cells, or benefit from a germband that participates in ventral cavity formation (Benton, 2018).”

The sentence has been rephrased as suggested and now reads: “Strong correlation of movements in serosa and yolk sac membranes indicated a tight coupling between the membranes in the phase of paused serosa spreading, which could be observed until the extending germband reached about the middle of the egg.”

6) Please state either in the Materials and methods or in the Results what temperature embryos were incubated at during live imaging, so that the reader can appropriately interpret the many values given in terms of absolute time in minutes.

We apologize for the omission of this information in the previous revision. The following sentence has now been added to the general microscopy section in Materials and methods: “In the absence of temperature-controlled stages, all time-lapse recordings have been acquired at room temperature. For both, confocal tissue recordings and initial in toto bright-field reference observations, room temperature was monitored continuously at 25°C.”

7) The final Discussion section on the mechanical aspects of the evolutionary origin of the Drosophila amnioserosa is succinct and appealing, but why assume a "gradual" loss of tissue decoupling?

As outlined above, we have rewritten the Discussion to provide a more balanced perspective on alternative hypotheses that have been proposed previously.